# Islet vascularization is regulated by primary endothelial cilia via VEGF-A-dependent signaling

Yan Xiong[1], M Julia Scerbo[2], Anett Seelig[2], Francesco Volta[2,3], Nils O'Brien[2], Andrea Dicker[1], Daniela Padula[2], Heiko Lickert[2,3,4], Jantje Mareike Gerdes[2,4†]*, Per-Olof Berggren[1†]

[1]The Rolf Luft Research Center for Diabetes and Endocrinology, Karolinska University Hospital L1, Stockholm, Sweden; [2]Institute for Diabetes and Regeneration Research, Helmholtz Zentrum München, Neuherberg, Germany; [3]Technical University Munich, Munich, Germany; [4]Deutsches Zentrum für Diabetesforschung, DZD, Munich, Germany

**Abstract** Islet vascularization is essential for intact islet function and glucose homeostasis. We have previously shown that primary cilia directly regulate insulin secretion. However, it remains unclear whether they are also implicated in islet vascularization. At eight weeks, murine $Bbs4^{-/-}$ islets show significantly lower intra-islet capillary density with enlarged diameters. Transplanted $Bbs4^{-/-}$ islets exhibit delayed re-vascularization and reduced vascular fenestration after engraftment, partially impairing vascular permeability and glucose delivery to β-cells. We identified primary cilia on endothelial cells as the underlying cause of this regulation, via the vascular endothelial growth factor-A (VEGF-A)/VEGF receptor 2 (VEGFR2) pathway. In vitro silencing of ciliary genes in endothelial cells disrupts VEGF-A/VEGFR2 internalization and downstream signaling. Consequently, key features of angiogenesis including proliferation and migration are attenuated in human $BBS4$ silenced endothelial cells. We conclude that endothelial cell primary cilia regulate islet vascularization and vascular barrier function via the VEGF-A/VEGFR2 signaling pathway.

*For correspondence:
jantje.gerdes@helmholtz-muenchen.de

[†]These authors contributed equally to this work

Competing interests: The authors declare that no competing interests exist.

## Introduction

Pancreatic islets rely on their morphologically unique vascular network for functional support and efficient delivery of oxygen and nutrients, as well as rapid dissemination of islet derived hormones. In addition, blood vessels are important to establish microenvironments or 'niches' required for stem or progenitor cell maintenance (*Brissova et al., 2014*). Proper islet re-vascularization has also been suggested to be a critical determinant in graft survival following islet transplantation in the treatment of *Type 1 Diabetes* (*Vaithilingam et al., 2008*; *Brissova et al., 2006*; *Brissova et al., 2004*). As the functional barrier between tissues and the blood stream, vascular endothelial cells in both central and peripheral organs of insulin action have been implicated in insulin resistance (*Konishi et al., 2017*) and diabetes etiology (*Goligorsky, 2017*). However, functional assessment of nutrient delivery and insulin disposal across the endothelial barrier has been difficult, since blood vessels are not preserved in vitro after islet isolation, while pancreas is difficult to access in situ with long-term, non-invasive live imaging techniques.

Anatomically, the exocrine pancreas is organized by branching ducts while the endocrine islets of Langerhans are scattered throughout the exocrine tissue and interconnected by blood vessels (*Cleaver and Dor, 2012*). During development, pancreatic bud and blood vessel formation are initiated at the same time and, although both endocrine and exocrine pancreatic tissue are derived from the same progenitors, vessel density in the exocrine portion of the organ is five times lower

compared to intra-islet vessel density after maturity. Moreover, islet capillaries have ten times more diaphragm-covered fenestrae than the exocrine pancreas- a distinctive feature that facilitates rapid exchange of fluids and small solutes between tissue and blood stream (*Kolka and Bergman, 2012*; *Henderson and Moss, 1985*). This suggests that specific local factors play a role in endocrine pancreatic vascularization.

One of the main players in islet vascularization is vascular endothelial growth factor-A (VEGF-A)/ VEGF receptor 2 (VEGFR2) dependent signaling. VEGF-A is produced by islet endocrine cells (both α- and β-cells) and binds to VEGFR2 on endothelial cells, which is crucial for the maintenance of a dense and leaky islet microvasculature (*Brissova et al., 2006*; *Lammert et al., 2003*; *Christofori et al., 1995*). While pancreas-specific VEGF-A ablation does not completely block islet vascularization, islet fenestration is crucially dependent on VEGF-A/VEGFR2 signaling. Delivery of nutrients to β-cells and disposal of insulin into the blood vessels is modulated by fenestrated blood vessels that are fine tuning the system to optimize glucose tolerance (*Lammert et al., 2003*). Constitutive, targeted deletion of *Vegfa* from β-cells at embryonic development revealed a critical role for VEGF-A dependent signals in islet vessel formation, maintenance and function (*Brissova et al., 2006*). β-cell-specific *Vegfa* deficient mice are severely glucose intolerant but show normal glucose response and insulin secretion in isolated islets, suggesting that either nutrients (such as glucose) are not delivered efficiently to the β-cells or insulin disposal into the blood stream is significantly hampered *in* vivo (*Brissova et al., 2006*; *Lammert et al., 2003*). In contrast, removal of *Vegfa* from mature β-cells at a later stage has a less severe impact on glucose tolerance although intra-islet capillary density is reduced by 50% (*Reinert et al., 2013*). This may suggest that, after islet maturation, local VEGF-A/VEGFR2 dependent signaling is less important for the maintenance of intra-islet capillaries and insulin disposal compared to during development. Changing the biomechanical properties such as actomyosin cortex tension of epithelial cells in developing islets by deletion of *Integrin-linked kinase* (*Ilk*) specifically blocks intra-islet capillary formation completely and leads to glucose intolerance. Instead, blood vessels accumulate in the islet periphery and form an envelope-like structure suggesting that biomechanics of epithelial tissues are important determinants of blood vessel formation (*Kragl et al., 2016*).

Primary cilia are present on roughly eighty percent of the cells in the adult mammalian organism (*Wheatley et al., 1996*). In metabolically active organs, they play a role in sensing metabolic signals and energy homeostasis (*Oh et al., 2015*; *Volta and Gerdes, 2017*). We have shown that β-cell cilia play a role in insulin signaling, insulin secretion and glucose homeostasis (*Volta et al., 2019*; *Gerdes et al., 2014*). Endothelial cells are also ciliated, and endothelial cilia have been implicated in flow-sensing and vascular hypertension, intracranial blood vessel formation, and atherosclerosis prevention (*Nauli et al., 2008*; *Nauli et al., 2011*; *Kallakuri et al., 2015*; *Dinsmore and Reiter, 2016*). Recent studies further unveiled a novel role of primary cilium in preventing vascular regression (*Vion et al., 2018*). Here we address the role of endothelial cilia in re-vascularization of transplanted islets and regulation of vascular barrier function.

## Results

### Intra-islet capillary density is reduced in *Bbs4*[-/-] pancreata

To test if primary cilia play a role in pancreatic islet vascularization, we characterized intra-islet capillaries in pancreatic cryosections of 2-month-old *Bbs4*[-/-] mice (N = 8). *Bbs4*[-/-] mice are a model of Bardet-Biedl Syndrome (OMIM #209900), a ciliopathy characterized mainly by polydactyly, renal and gonadal malformations and truncal obesity. *Bbs4* encodes a protein that is a component of the BBSome complex, a protein super complex thought to be involved in the sorting of membrane proteins to and from the ciliary compartment (*Nachury et al., 2007*; *Liew et al., 2014*). While *Bbs4*[-/-] mice form primary cilia, they are not fully functional (*Mykytyn et al., 2004*). The symptoms show good overlap with what has been observed in BBS patients and include obesity, male sterility, and impaired glucose handling. Platelet endothelial cell adhesion molecule (PECAM-1) immunofluorescence, as a marker of endothelial cells, revealed 35% reduced intra-islet capillary density compared to wildtype (wt) littermate controls (*Figure 1A,B*). To calculate intra-islet capillary density, the relative PECAM-1 positive volume was normalized to insulin-positive islet volume (*Figure 1B*, p=0.0019). In addition, the average vessel diameter was increased by 18% compared to that in wt controls

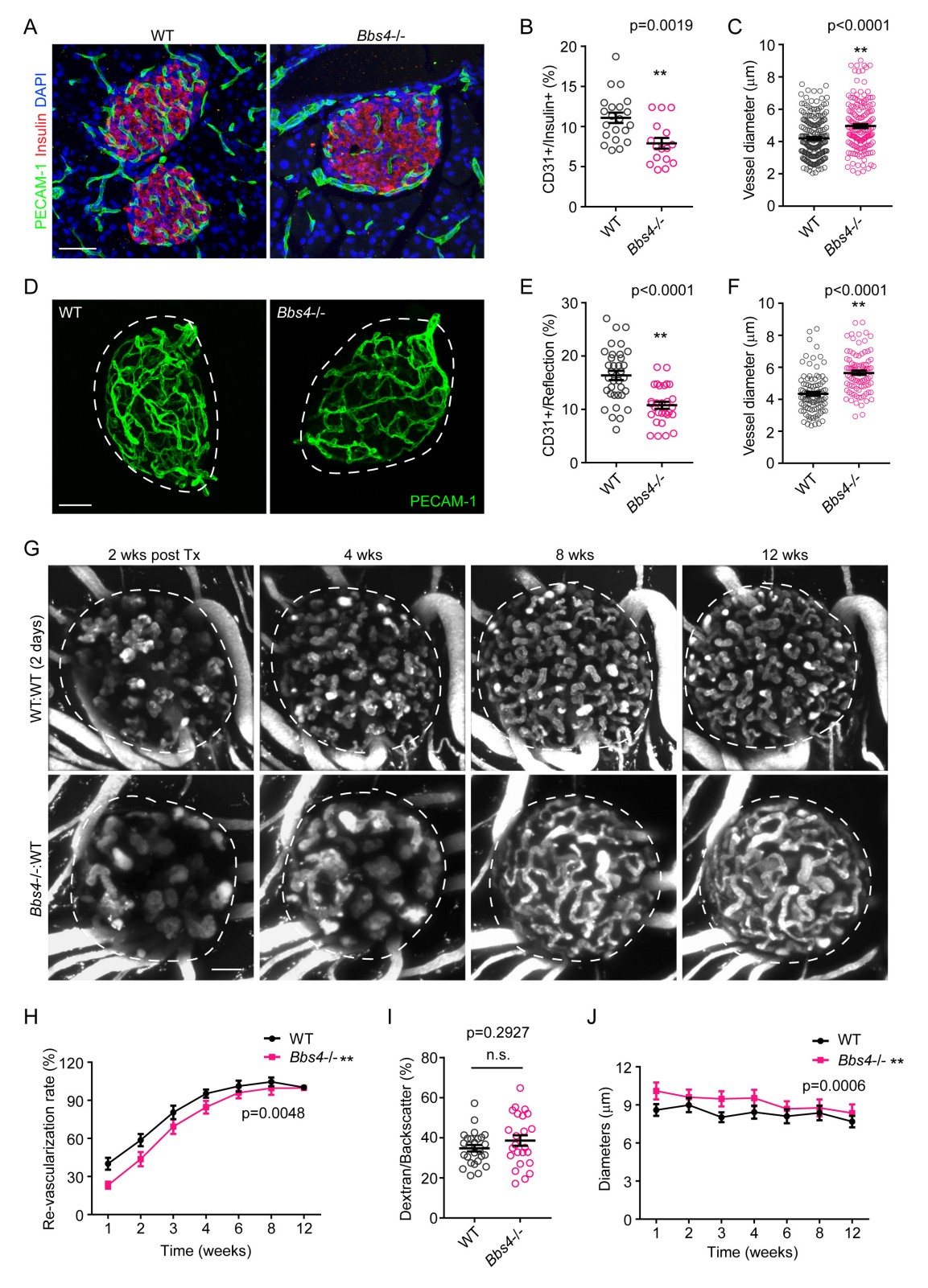

**Figure 1.** *Bbs4*$^{-/-}$ islets show delayed vascularization and enlarged capillary diameter in the pancreas and the anterior chamber of the eye upon transplantation. (**A**) Immuno-fluorescence staining of pancreatic sections from 2-month-old wt and *Bbs4*$^{-/-}$ mice showing islets (insulin, red) and intra-islet capillaries (PECAM-1, green). (**B–C**) Quantification of relative intra-islet PECAM-1 positive volume, normalized to insulin-positive volume (**B**) and average intra-islet capillary diameters (**C**) in wt and *Bbs4*$^{-/-}$ pancreatic sections. Individual data points are shown, **p<0.01, n = 8 for animals and n = 2–

*Figure 1 continued on next page*

*Figure 1 continued*

3 islets per animal. (**D**) Immuno-fluorescence staining of freshly isolated and fixed pancreatic islets from 2-month-old wt and *Bbs4^-/-^* mice, showing PECAM-1 (green) labeled islet capillaries. (**E–F**) Quantification of relative PECAM-1 positive volume within each islet, normalized to islet volume estimated by backscatter signal (**E**) and average capillary diameters (**F**) in wt and *Bbs4^-/-^* islets. Individual data points are shown, **p<0.01 by Mann-Whitney test, n = 3 for animals and n = 8–12 islets per animal. (**G**) Re-vascularization of 2 day-cultivated wt (upper) and *Bbs4^-/-^* (lower) islets in wt recipient eyes at 2-, 4-, 8- and 12-weeks post-transplantation, visualized by intravenous injection of Texas Red-conjugated dextran. (**H**) Quantification of re-vascularization rates of wt and *Bbs4^-/-^* islet grafts in wt recipients. Results are shown as mean ± SEM. (**I**) Relative vascular density of wt and *Bbs4^-/-^* islet grafts at the end of 12 weeks post-transplantation. Individual data points are shown, n.s. means not significant by Mann-Whitney test. (**J**) Average diameters of newly formed capillaries in wt and *Bbs4^-/-^* islet grafts in wt recipients. Results are shown as mean ± SEM, *p<0.05, **p<0.01 by two-way-ANOVA, n = 6 for animals and n = 4–8 islets per animal. Islets were encircled by dashed lines. The same is for all the other figures. Scale bars: 50 μm.

The online version of this article includes the following figure supplement(s) for figure 1:

**Figure supplement 1.** Pericyte coverage of intra-islet capillaries and capillary density in exocrine pancreas are not affected in *Bbs4^-/-^* mice.
**Figure supplement 2.** Number of donor endothelial cells participating in islet re-vascularization decreases under prolonged cultivation.

---

(*Figure 1C*, p<0.0001). Because tissue integrity is often compromised in cryopreserved samples, we corroborated our results by whole mount staining and imaging of freshly isolated islets of 2-month-old mice (*Figure 1D–F*). In good agreement with our previous observations, capillary density was reduced by 34% and vessel diameter increased by 20% in islets of *Bbs4^-/-^* mice (p<0.0001 for both). Of note, we observed no change in pericyte coverage of intra-islet capillaries based on Neuron-glial 2 (NG2) immunofluorescence (*Figure 1—figure supplement 1A and B*, p=0.7045). In 4-month-old *Bbs4^-/-^* mice, however, the difference in intra-islet capillary density and diameter in *Bbs4^-/-^* mice approximated that of wt littermates (*Figure 1—figure supplement 1C–1E*, p=0.0676 and 0.0736 respectively). There is no evidence of significant changes in β-cell mass or insulin content in *Bbs4^-/-^* mice, and the observation of only slightly but not statistically significantly larger islets in *Bbs4^-/-^* mice at 4 weeks of age rules out effects of islet size on vessel density (*Gerdes et al., 2014*). These dynamics implicate that primary cilia and centrosome/basal body integrity play a role during the vascularization of pancreatic islets. Importantly, there was no detectable difference in vascularization of the exocrine portion of the pancreas at two months of age (*Figure 1—figure supplement 1F*). Therefore, *Bbs4* function seems to be mostly restricted to the endocrine pancreas and not relevant for the exocrine compartment.

## Bbs4^-/- endothelial cells exhibits blunted angiogenic response during islet re-vascularization

To test if cilia play a role in intra-islet capillary formation during engraftment of transplanted islets, we transplanted murine islets into the anterior chamber of the eye (ACE) (*Speier et al., 2008*). This approach allows for longitudinal and non-invasive in vivo monitoring of islet engraftment and re-vascularization. We and others have previously shown that endothelial cells disappear over prolonged periods of cultivation of isolated islets. Immediately after isolation, endothelial cells maintain the intra-islet capillary network. After two days in culture, endothelial cell clusters remain, whereas seven days post-isolation, intra-islet endothelial cells are generally lost (*Figure 1—figure supplement 2A*). Thus, when transplanted shortly after isolation, a significant number of endothelial cells from the donor survives and contributes to revascularization of the islet graft (*Brissova et al., 2004*; *Linn et al., 2003*; *Nyqvist et al., 2005*). Importantly, both islet cells and endothelial cells are ciliated (*Nauli et al., 2008*; *Dinsmore and Reiter, 2016*; *Hughes et al., 2020*; *Ma and Zhou, 2020*). Therefore, to determine whether the role of primary ciliary/centrosomal function in islet endothelial or endocrine cells is underlying the reduction in intra-islet capillary density in *Bbs4^-/-^* mice, we capitalized on the temporal differences in the ratio of donor or recipient endothelial cells in regenerated islet capillary network (*Figure 1—figure supplement 2B*).

In one experimental setting, islets isolated from *Bbs4^-/-^* and wt littermate controls were cultivated for two days before transplantation into the ACE of wt B6 albino recipients (*Figure 1G*, *Figure 1—figure supplement 2B*, upper panel). We determined the rate of re-vascularization by intravenous (i. v.) injection of fluorescently labeled 70 KDa dextran once a week for a total of twelve weeks (*Figure 1G*). The percentage of dextran-labeled vascular structure in total islet volume (based on backscatter signal; *Ilegems et al., 2015*) was used as a measure of islet vascularity, and re-vascularization rate of each islet was calculated by normalizing its dextran-labeled volume at each individual

week to the volume at twelve weeks. Two weeks after transplantation, $Bbs4^{-/-}$ islets showed significantly less re-vascularization compared to wt controls (*Figure 1H*). Until four weeks after transplantation, re-vascularization remains lower than that of wt islets (*Figure 1H*). Twelve weeks post-transplantation, relative intra-islet vascular density is similar in both $Bbs4^{-/-}$ and wt islets (*Figure 1I*). Morphologically, the newly formed blood vessels also differ among the two different genotypes during the first few weeks of engraftment. The vessel diameter is greater in $Bbs4^{-/-}$ islets, suggesting that there are fewer, wider islet capillaries in these islets compared to the wt controls during the first four weeks of engraftment (*Figure 1J*), similar to the results obtained from cryosections of young $Bbs4^{-/-}$ animals.

In the other experimental setting, islets were kept in culture for seven days after isolation before being transplanted to the ACE of wt recipients. This treatment ablates endothelial cells in the donor tissue and favors almost exclusively revascularization by the recipients' vascular system (*Figure 1—figure supplement 2B*, lower panel). We did not observe significant differences in re-vascularization rate over time or islet vascular density at the twelve-week endpoint (*Figure 2A–C*). In addition, vessel morphology and diameter were similar in $Bbs4^{-/-}$ and wt control islets (*Figure 2D*). Importantly, under these experimental conditions, the majority of donor endothelial cells were lost during the seven-day cultivation. In conclusion, our findings suggest that $Bbs4^{-/-}$ endothelial cells underlie the blunted angiogenic response, thus impeding islet re-vascularization.

To verify our conclusion, we reversed the transplantation strategy by transplanting wt islets to $Bbs4^{-/-}$ recipients either directly after isolation or after keeping them in culture for two and seven days respectively (*Figure 3*). To minimize confounding effects of diabetes on islet engraftment, we transplanted islets into the eyes of four-month-old $Bbs4^{-/-}$ mice that were obese and glucose intolerant, but not overtly diabetic yet (*Figure 3A–C*). Based on the apparent role of endothelial primary cilia in the angiogenic response, we expected more severe impairment in re-vascularization compared to wt recipients. Indeed, we observed significantly slower re-vascularization in this transplantation scheme and the rates correlated with the time islets were kept in culture (*Figure 3D and E*). The re-vascularization rate of wt islets transplanted to the ACE of $Bbs4^{-/-}$ mice seven days post-isolation was significantly lower than that of wt islets two days post-isolation (p=0.0053), and both were significantly lower than that of wt islets transplanted to wt ACE (p=0.0002 for two-day group and p<0.0001 for seven-day group respectively). Interestingly, intra-islet vessel density after completion of re-vascularization was not significantly lower than that of wt islets transplanted into wt recipients at the twelve-week endpoint (*Figure 3F*). The diameter of intra-islet capillaries, however, was markedly wider in wt islets transplanted to $Bbs4^{-/-}$ recipients after seven days in culture than the other two groups (*Figure 3G*).

To further distinguish donor and recipient endothelial cells in ACE islet grafts and trace their dynamics during engraftment, we used a tdTomato reporter mouse line (Ai14) that carries a *loxP* flanked STOP codon preventing expression of a CAG-driven red fluorescent protein (*tdTomato*) in the *Rosa26* locus. Crossing the floxed allele to an endothelial specific transgenic *Cdh5-Cre* mouse line (*Chen et al., 2009*) rendered all VE-Cadherin expressing-endothelial cells labeled with tdTomato (*Cdh5-tdT*). Directly after isolation, red fluorescence efficiently labels the intact intra-islet capillary network, which is demonstrated by PECAM-1 staining (*Figure 3—figure supplement 1A*). As expected, this network is partially preserved when transplanted shortly after isolation, and forms a mosaic neo-vasculature in islet grafts together with recipient endothelial cells (*Figure 3—figure supplement 1B*, left panel). Twelve weeks post-transplantation, we quantified the surface area ratio of tdTomato-positive structures to the lumen of the blood vessels as marked by fluorescein-labeled dextran and found it to be 41.7% in wt and 45.9% in $Bbs4^{-/-}$ islets transplanted two days after culture. This is in good agreement with another study, in which freshly transplanted islets were under the kidney capsule; here, the authors observed that donor endothelial cells constituted 40 ± 3% of the newly formed vasculature between three to five weeks post-transplantation (*Brissova et al., 2004*). On the contrary, only little tdTomato fluorescence was detected in the *Cdh-tdT* islets cultivated over seven days, suggesting that re-vascularization is mostly derived from non-fluorescent recipient endothelial cells (*Figure 3—figure supplement 1B*, right panel). To confirm previous findings and better understand the relative contribution of donor versus recipient endothelial cells during islet graft vascularization, we transplanted Ai14;*Cdh5-Cre* animals with islets isolated from $Bbs4^{-/-}$ and wt littermate controls which were cultivated for two days and monitored their re-vascularization (*Figure 3—figure supplement 1C*). While delayed re-vascularization and altered vascular

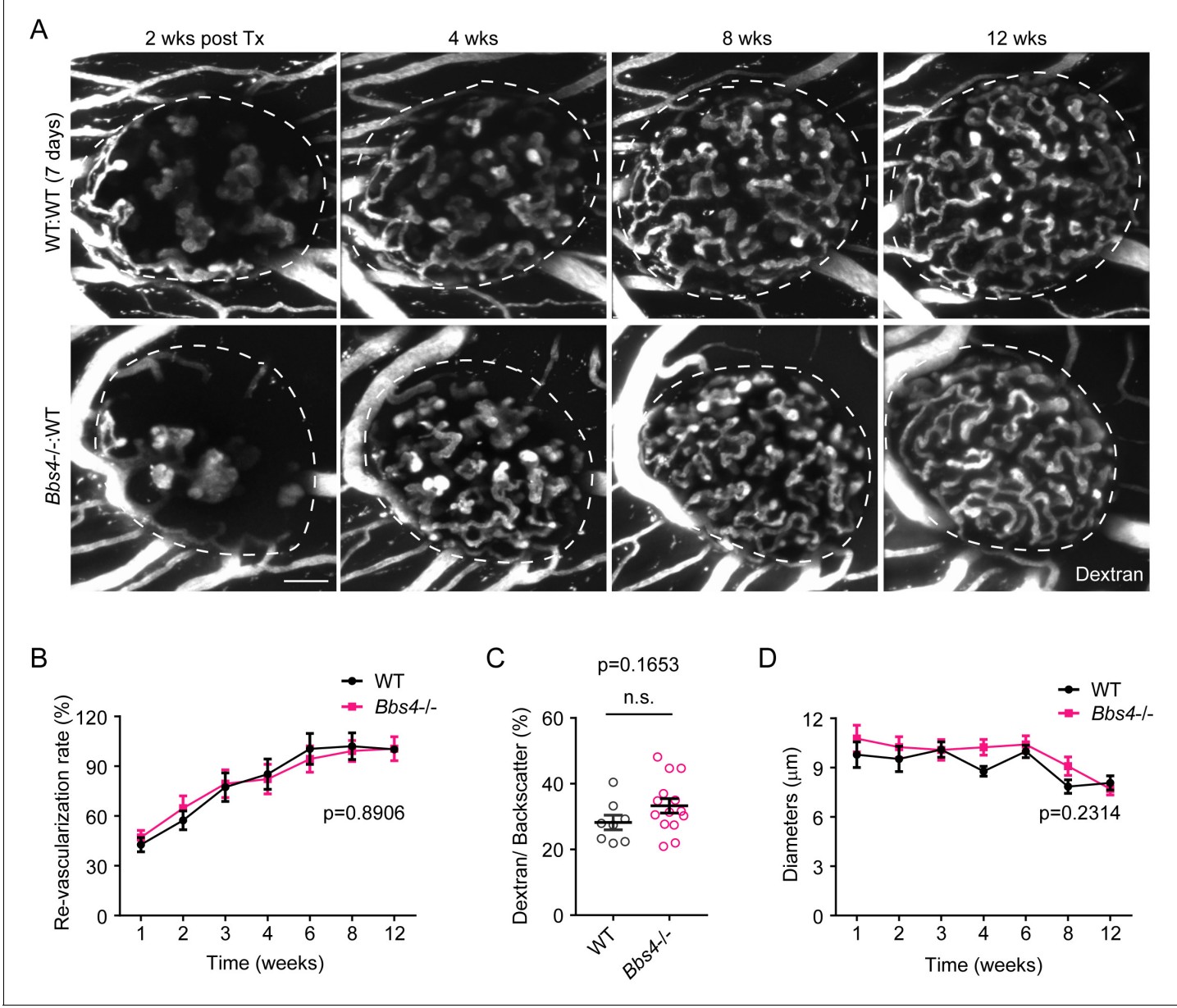

**Figure 2.** *Bbs4*[-/-] islets after prolonged culture exhibit normal re-vascularization patterns in wt recipient eyes. (**A**) Re-vascularization of 7-day-cultivated wt (upper) and *Bbs4*[-/-] (lower) islets in wt recipient eyes at 2-, 4-, 8- and 12-weeks post-transplantation, visualized by intravenous injection of Texas Red-conjugated dextran. Donor islets have been cultured for 7 days prior to transplantation. (**B**) Quantification of re-vascularization rates of wt and *Bbs4*[-/-] islet grafts in wt recipients. Results are mean ± S.E.M. (**C**) Relative vascular density of wt and *Bbs4*[-/-] islet grafts at the end of 12 weeks post-transplantation. Individual data points are shown, n.s. means not significant by Mann-Whitney test. (**D**) Average diameters of newly formed capillaries in wt and *Bbs4*[-/-] islet grafts in wt recipients. Results are mean ± S.E.M., n = 4 for animals and n = 2–6 islets per animal. Scale bar: 50 μm.

morphology of *Bbs4*[-/-] islets was observed as before in wt recipients (*Figure 3—figure supplement 1D and E*, *Figure 1H and J*), the ratio of recipient-derived tdTomato-positive endothelial cells stayed significantly higher in *Bbs4*[-/-] islets during the first four weeks of engraftment (*Figure 3—figure supplement 1F*). Taken together, these results support the conclusion that *Bbs4*[-/-] endothelial cells are indeed less responsive to the angiogenic signals than wt endothelial cells.

## Primary cilia of endothelial cells regulate islet re-vascularization

Because Bbs4 protein might have additional roles in cellular processes unrelated to the basal body, we used two additional mutant mouse models. *Pitchfork* (*Pifo*[-/-]) mice have dysfunctional cilia that

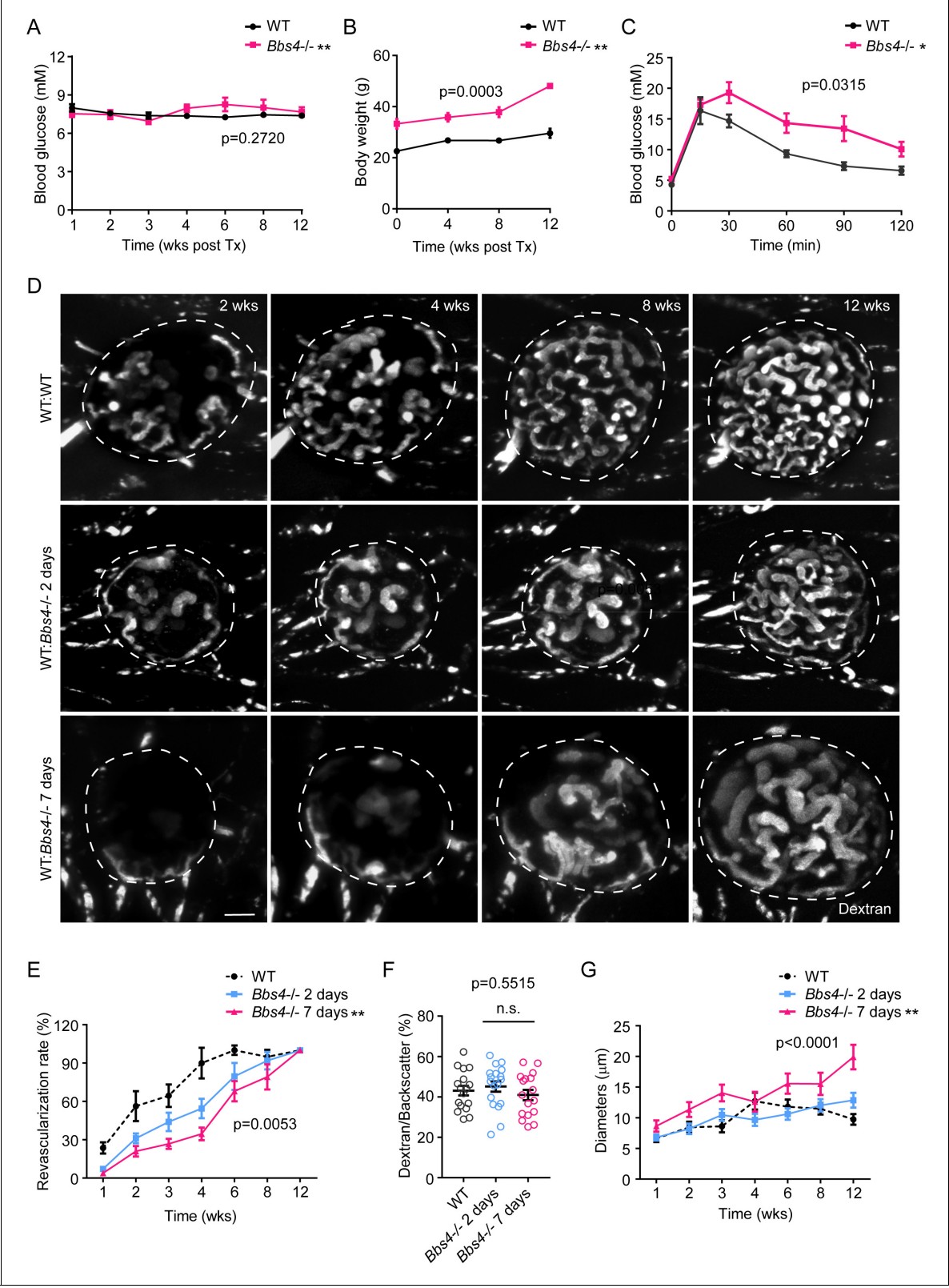

**Figure 3.** Wildtype islet grafts transplanted into *Bbs4*[-/-] recipient eyes show stronger impairment in re-vascularization. (A) Non-fasting glycemic levels in *Bbs4*[-/-] mice during islet engraftment. (B) Body weight measurements of *Bbs4*[-/-] mice during engraftment. (C) Intraperitoneal glucose tolerance test 12 weeks post-transplantation. Results are mean ± S.E.M. *p<0.05, **p<0.01 by two-way-ANOVA, n = 5. (D) Re-vascularization of wt islets (upper) after 2 (middle) or 7 days (lower) of culture in wt or *Bbs4*[-/-] recipients at 2-, 4-, 8- and 12-weeks post-transplantation, visualized by intravenous injection of Texas

*Figure 3 continued on next page*

Figure 3 continued

Red-conjugated dextran. (**E**) Quantification of re-vascularization rates of wt islet grafts. Results are mean ± S.E.M. (**F**) Relative vascular density of wt islet grafts in different recipients at the end of 12 weeks post-transplantation. Individual data points are shown, n.s. means not significant by one-way-ANOVA. (**G**) Average diameters of newly formed capillaries in wt islet grafts. Results are mean ± S.E.M. Comparisons were made between 2 day and 7 day groups by two-way-ANOVA, *p<0.05, **p<0.01. n = 4 for animals and n = 2–4 islets per animal for wt recipients. n = 5 for animals and n = 1–4 islets per animal for *Bbs4*$^{-/-}$ recipients. Scale bar: 50 μm.

The online version of this article includes the following figure supplement(s) for figure 3:

**Figure supplement 1.** *Bbs4*$^{-/-}$ endothelial cells are less responsive to the angiogenic signals during islet re-vascularization.

**Figure supplement 2.** *Pifo-/-*mice exhibit similar islet vascular phenotypes as *Bbs4*$^{-/-}$ mice.

cannot be properly disassembled (*Kinzel et al., 2010*). PIFO does not localize to the ciliary axoneme but accumulates at the base of the cilium during cilia disassembly and interacts with ciliary targeting complexes. Laterality defects in *Pifo*$^{-/-}$ mice suggest that ciliary signaling is impaired in these animals. We determined intra-islet vascular density in 3-month-old *Pifo*$^{-/-}$mice. PECAM-1 staining revealed 19% reduction in intra-islet vascular density and 38% increase in vessel diameter in *Pifo*$^{-/-}$ mice compared to littermate controls, similar to our observation in *Bbs4*$^{-/-}$ islets (*Figure 3—figure supplement 2A–2C*, p=0.0421 and p=0.0032 respectively). Transplantation of *Pifo*$^{-/-}$ islets into wt ACE showed a delay in relative re-vascularization compared to wt islets, also similar to what we observed for *Bbs4*$^{-/-}$ mice (*Figure 3—figure supplement 2D–2G*). Although the effects on re-vascularization were less severe than those in *Bbs4*$^{-/-}$ islets, the findings support a role for primary cilia in islet vascularization and the angiogenic response in islet engraftment.

To further clarify the contributions of β-cell vs. endothelial cilia to the vascularization phenotype and confirm the findings from *Bbs4*$^{-/-}$ mice, we specifically ablated *Intraflagellar transport 88* (*Ift88*), a core ciliary protein, in β-cells by crossing a floxed *Ift88* mouse line to an inducible cyclic recombinase line under the control of *Pdx1-* promoter. This generates a β-cell-specific, inducible ciliary knockout mouse (*Ift88*$^{loxP/loxP}$; *Pdx1-CreER*) with an efficiency of 50%, that we will refer to as βICKO (pronounced BICKO) from here on (*Volta et al., 2019*; *Figure 4A*). Ift88 is essential to ciliary maintenance and formation, and cells depleted of Ift88 do not have ciliary axonemes (*Davenport et al., 2007*), although the early stages of ciliogenesis such as vesicle docking to centrosomes are still completed (*Schmidt et al., 2012*). Recombination was induced between postnatal day 25 and 35, and we quantified intra-islet vascular density in Tamoxifen-treated βICKO mice six weeks after induction. We found no observable difference between induced βICKO mice and oil-treated controls (*Figure 4B and C*). In addition, we isolated βICKO islets two months after Tamoxifen treatment and transplanted them into wt mice after overnight culture. We did not observe differences in the re-vascularization rate between vehicle and Tamoxifen-treated βICKO islets (*Figure 4D–G*). Overall, our observations strongly suggest that cilia in endothelial cells regulate re-vascularization during islet engraftment.

## Intra-islet vasculature plays a role in glucose metabolism

Besides a significantly higher vascular density than exocrine pancreas, another hallmark of intra-islet capillary are fenestrae as gateways for substance exchange between islets and the blood stream (*Henderson and Moss, 1985*). Thus, we further examined the barrier function of *Bbs4*$^{-/-}$ islet vessels four months after engraftment, when the intra-islet vascular density is comparable with wt islets. As previously reported, in isolated islets or *Bbs4* depleted MIN6m9 cells, we did not observe a difference in β-cell glucose uptake or glucose metabolism. However, the role of *Bbs4* in islet endothelial cell barrier function remains uncharacterized. Blinded analysis of electron micrographs showed significantly reduced fenestration in the capillaries of *Bbs4*$^{-/-}$ compared to wt islet grafts in the ACE four months after transplantation (*Figure 5A and B*, 27% lower, p=0.0006). Although the basement membrane seems slightly thicker in *Bbs4*$^{-/-}$ islets, we did not observe significant thickening of this structure (*Figure 5C*, p=0.1414). To test if reduced fenestration could affect the permeability and thus barrier function of *Bbs4*$^{-/-}$ islet capillaries, we intravenously injected 40 kDa fluorescently labeled dextran and measured the diffusion of dextran-related fluorescence outside the islet graft over time (inside the dashed area not including the part of vessel, *Figure 5D*). Fluorescently labeled dextran of different sizes were tested prior to the experiments, and we chose 40 KDa dextran due to its specific

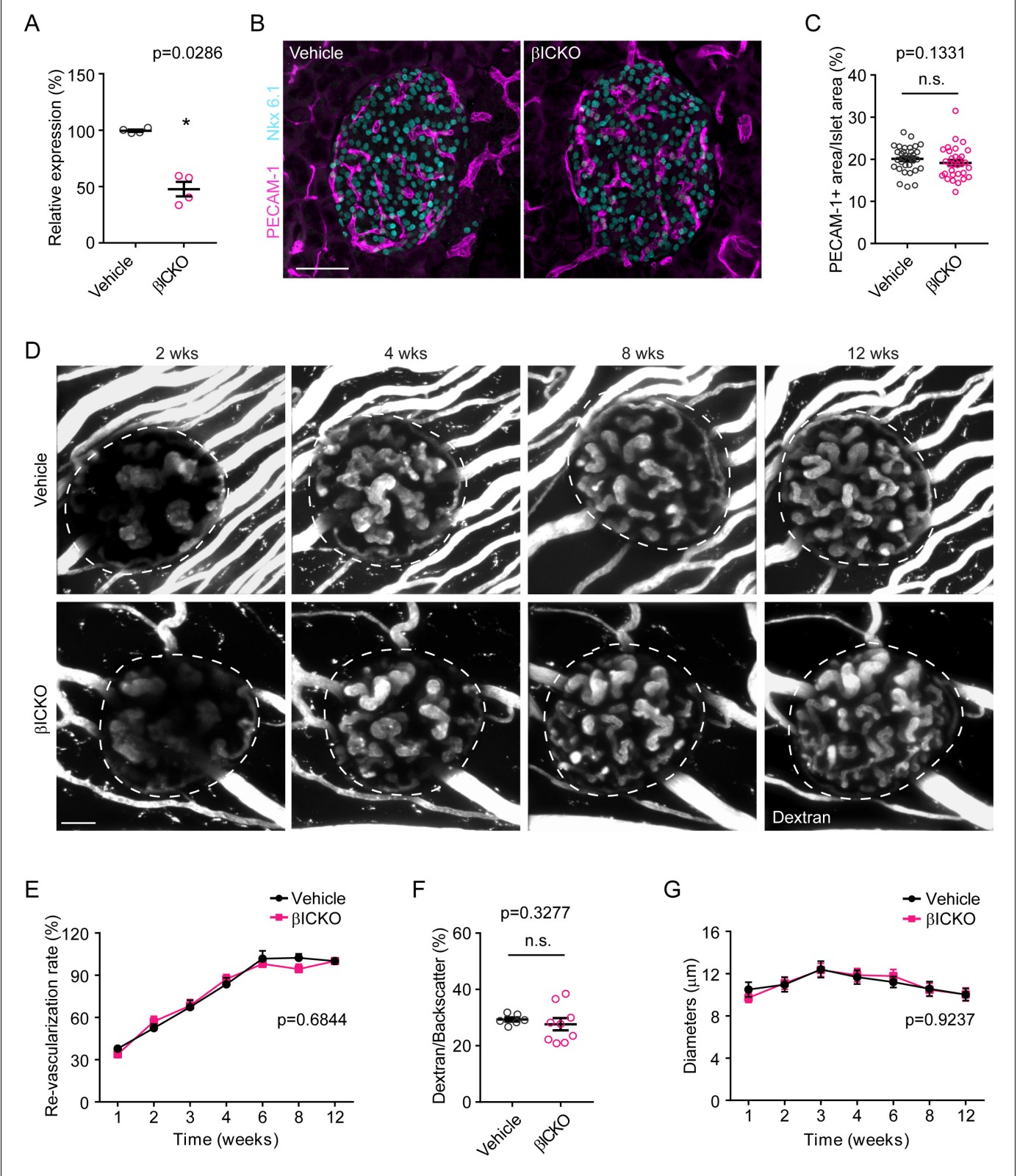

**Figure 4.** βICKO mice display no visible islet vascular phenotypes. (**A**) Efficiency of *Ift88* silencing by tamoxifen induction. Individual data points are shown, **p<0.01 by Mann-Whitney test, n = 4 for animals. (**B**) Immuno-fluorescence staining of pancreatic sections from 2-month-old control (vehicle) and βICKO mice, showing islets (Nkx 6.1, cyan) and intra-islet capillaries (PECAM-1, magenta). (**C**) Quantification of relative intra-islet PECAM-1 positive area, normalized to islet area in control and βICKO pancreatic sections. Individual data points are shown, n.s. means not significant by Mann-Whitney

*Figure 4 continued on next page*

Figure 4 continued

test, n = 6 for animals and n = 5–6 islets per animal. (D) Re-vascularization of overnight-cultured control (upper) and βICKO (lower) islets in wt recipient eyes at 2-, 4-, 8- and 12-weeks post-transplantation, visualized by intravenous injection of Texas Red-conjugated dextran. (E) Quantification of re-vascularization rates of control and βICKO islet grafts in wt recipients. Results are mean ± S.E.M. n = 6 for animals and n = 6–8 islets per animal. (F) Relative vascular density of control and βICKO islet grafts at the end of 12 weeks post-transplantation. Individual data points are shown. n = 3 for animals and n = 2–3 islets per animal. (G) Average diameters of newly formed capillaries in control and βICKO islet grafts in wt recipients. Results are mean ± S.E.M. and n.s. means not significant by Mann-Whitney test. n = 6 for animals and n = 6–8 islets per animal. Scale bars: 50 µm.

leakage through fenestrated islet vessels instead of surrounding iris vessels (*Figure 5—figure supplement 1*). During the time of the experiments, fluorescence intensity immediately increases outside the islet graft followed by a plateau that is established at comparable times after dextran injection due to the active drainage of aqueous humor. Fluorescence intensity is normalized to islet size for better comparison. Importantly, both the initial rate and the plateau are decreased in *Bbs4*$^{-/-}$ islet grafts (*Figure 5E*, p=0.0478). Simulation of this process also supports different kinetics of dye leakage (*Figure 5F*), suggesting that the delivery of medium sized molecules out of the blood stream is less efficient in *Bbs4*$^{-/-}$ intra islet capillaries. Nutrients such as glucose or fatty acids are small molecules that often rely on active transport across cell membranes. But in the case of highly fenestrated capillaries, solutes including small molecules, ions and hormones pass freely through the pores between the blood and luminal surface. To test if glucose delivery to islet cells was affected by the ultrastructural changes in islet capillaries, we injected fluorescently labeled glucose analog (2-NBDG) into wt animals transplanted with *Bbs4*$^{-/-}$ and wt islets. 2-NBDG is rapidly transported into islet cells from intra-islet capillaries and simultaneously leaked into the surrounding aqueous humor from iris vessels. We followed 2-NBDG uptake by measuring fluorescence intensity originating from islet cells over time, and normalized it to the fluorescence intensity originating from the aqueous humor (*Figure 5G*), since the 2-NBDG leaking from iris vessels is determined by the recipient animal and not by the individual eye. Importantly, wt islet cells internalized significantly more 2-NBDG than *Bbs4*$^{-/-}$ islet cells under the same period of time (*Figure 5H*, p=0.0118). This result points to a critical role of ciliary/basal body proteins in maintaining islet capillary fenestration with implications for the delivery of glucose and potentially other nutrients to islet cells. To investigate the physiological relevance of our findings, we utilized our transplantation model in combination with *Bbs4*$^{-/-}$ or wt islets to better understand how reduced intra-islet capillary density and permeability may affect islet output and glucose tolerance. It was previously shown that 100 islets (diameter between 150 and 200 µm) transplanted into the ACE of chemically induced diabetic mice were sufficient to revert blood glucose concentration to normoglycemic levels. This amount of pancreatic islets is defined as marginal islet mass (*Rodriguez-Diaz et al., 2018*). C57Bl/6 J mice were treated with strepotozotocin and subsequently transplanted with marginal islet mass of wt or *Bbs4*$^{-/-}$ islets. Four weeks after transplantation, mice transplanted with *Bbs4*$^{-/-}$ islets were glucose intolerant compared to those mice transplanted with wt islets (*Figure 5I*). Eight weeks after transplantation, mice showed impaired glucose tolerance and, 12 weeks post-transplantation, only mildly impaired glucose handling compared to those transplanted with wt islets (*Figure 5J and K*). Overall, the phenotype is ameliorated over time, which is consistent with our previous observations in re-vascularization (*Figures 1* and *2*), suggesting that a lack of capillaries in *Bbs4*$^{-/-}$ islets indeed negatively affects whole body glucose homeostasis. Glucose handling is mildly impaired by the end of 12 weeks, which is probably caused by impaired vascular barrier function in combination with defective 1$^{st}$ phase insulin secretion (*Gerdes et al., 2014*).

## Impaired VEGFA-VEGFR2 dependent signaling in *Bbs4*-depleted endothelial cells

We have observed a role for endothelial cilia in the islet vascularization process; the signaling pathways involved in this process, however, remained elusive. To complement the in vivo data and to better understand the cell-autonomous role of ciliary/basal body proteins in endothelial cells, we used a murine pancreatic endothelial cell line, MS-1. Suppressing *Bbs4* by RNA interference reduces *Bbs4* expression up to two-fold (*Figure 6—figure supplement 1A*). Expression analysis of key angiogenic factors revealed no observable changes in Tie1, Tie2, endothelial Nitric oxide synthase (eNOS) or Notch ligand Delta-4 (Dll4) (*Figure 6A*). However, VEGFR2, Ephrin A1 and B2 are downregulated

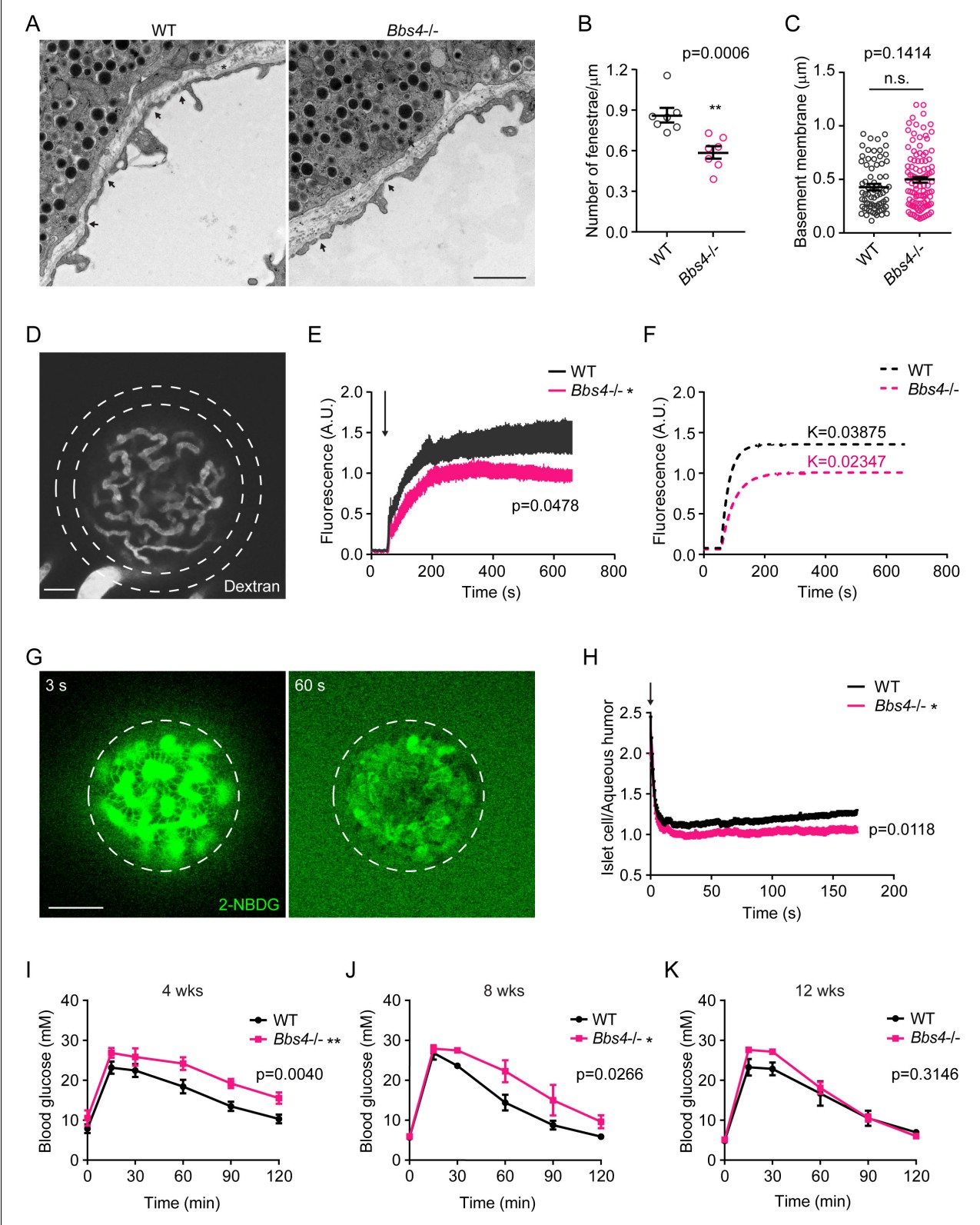

**Figure 5.** Dysfunctional intra-islet vasculature in *Bbs4*[-/-] islets undermines glucose metabolism. (**A**) Electron microscopic images of wt and *Bbs4*[-/-] islet graft dissected from wt recipient eyes, showing fenestrated islet capillaries (arrows) and basement membrane (asterisks). Scale bar: 1 μm. (**B–C**) Quantification of average fenestrae density (**B**) and basement membrane thickness (**C**) of the capillaries in wt and *Bbs4*[-/-] islet grafts. Individual data points are shown, **p<0.01, n.s. means not significant by Mann-Whitney test, n = 7 for animals, and n = 1–2 for islets. (**D**) Representative image showing

*Figure 5 continued on next page*

*Figure 5 continued*

leakage of 40 KDa FITC-conjugated dextran from wt islet grafts in mouse eyes at 1 min after injection. Scale bar: 50 μm. (**E–F**) Quantification of FITC fluorescence intensity in the region circles by the dashed lines outside wt and *Bbs4⁻ᐟ⁻*islet grafts (**E**) and simulated curves showing different kinetics (**F**). Arrow indicates injection time point. Results are mean ± S.E.M. *p<0.05 by two-way-ANOVA, n = 12 for animals. (**G**) Representative image showing 2-NBDG leakage from wt islet vasculature and uptake by islet cells in vivo. Times points are 3 s (left) and 60 s (right) after injection. Scale bar: 50 μm. (**H**) Real-time ratio of 2-NBDG fluorescence intensity in islet cells of wt and *Bbs4⁻ᐟ⁻* islet grafts to aqueous humor. Arrow indicates injection time point. Results are mean ± S.E.M. *p<0.05, n = 8 for animals and n = 1–2 for islets. (**I–K**) Intraperitoneal glucose tolerance test of wt recipient mice which were transplanted with wt or *Bbs4⁻ᐟ⁻* islets, at 4 weeks (**I**), 8 weeks (**J**) and 12 weeks (**K**) post-transplantation. Results are mean ± S.E.M. *p<0.05, **p<0.01 by two-way-ANOVA, n = 6 for animals.

The online version of this article includes the following figure supplement(s) for figure 5:

**Figure supplement 1.** Dextran leakage kinetics from the capillaries of islet grafts or the iris vessels in mouse eyes.

by 20% when MS-1 cells are depleted of Bbs4. Interestingly, Bbs4 depletion has no effect on PECAM-1 expression and therefore does not change the characteristic properties of this endothelial cell line. Of note, intra-islet capillary density and fenestration is primarily maintained by VEGFA-VEGFR2-signaling (*Lammert et al., 2003*; *Esser et al., 1998*; *Roberts and Palade, 1995*).

Angiogenesis is not exclusively regulated on the transcriptional level but also relies on a complex network of signaling cascades. Therefore, we tested if VEGFR2-dependent phospho-activation of downstream signaling molecules, including Phospholipase Cγ1 (PLCγ1), Protein Kinase B (Akt) and Mitogen-activated protein kinase (MAPK 42/44) were affected. We used primary human endothelial cells HDMEC stably transduced with lentivirus encoding shRNA targeting *BBS4*, and tested before (t = 0), seven and fifteen minutes after VEGF-A addition (*Figure 6B–D*, *Figure 6—figure supplement 1B and C*). In control cells (scrambled), VEGFR2 Tyr 1175 phosphorylation is markedly increased at seven minutes (4.3-fold) and sustained until 15 min (3.4-fold) after stimulation (*Figure 6B and D*). By comparison, pTyr1175 levels increased minimally in *BBS4*-depleted HDMECs (2-fold and 1.8-fold, respectively, *Figure 6B and D*). At the same time, total levels of VEGFR2 are also lowered in absence of Bbs4 by 20% (*Figure 6B*, *Figure 6—figure supplement 1B*, p=0.0101). In addition, we found that EphrinB2 protein level is slightly reduced in *BBS4*-depleted HDMECs (*Figure 6B*, *Figure 6—figure supplements 1B* and 5%, p=0.0051), corroborating our previous finding of decreased gene expression (*Figure 6A*). While PLCγ1 phosphorylation peaks at seven minutes and decreases 15 min after stimulation in control cells, loss of Bbs4 leads to a 40% attenuation in PLCγ1 phosphorylation at seven minutes (*Figure 6C*, *Figure 6—figure supplement 1C*, p=0.0336). Similarly, pAkt and pMAPK 42/44 were significantly reduced in *BBS4*-depleted cells seven and fifteen minutes after VEGF-A addition (*Figure 6C*, *Figure 6—figure supplement 1C*, p=0.0222 and p=0.0010, respectively). These in vitro experiments are suggestive but do not necessarily reflect in vivo processes of islet vascularization. Therefore, we further validated these results in freshly isolated islets from wt and *Bbs4⁻ᐟ⁻* animals, which still contain endothelial cells (*Figure 6—figure supplement 1D*). In unstimulated islet cells, pAkt and pMAPK baseline levels did not differ between wt and *Bbs4⁻ᐟ⁻* cells. Ten minutes after addition of 100 ng/ml VEGF-A, pAkt levels increased 2.2-fold in wt but only 1.4-fold in *Bbs4⁻ᐟ⁻* cells. Likewise, pMAPK levels increased 3-fold in wt but only 1.5-fold in *Bbs4⁻ᐟ⁻* cells, demonstrating that VEGF-A triggered downstream signaling is not sufficiently activated in these cells (*Figure 6—figure supplement 1D and E*). Importantly, similar experiments on wt islets cultured for seven days to diminish intra-islet endothelial cells showed markedly lower response to VEGF-A stimulation; therefore, VEGF-A/VEGFR2 dependent signaling cascades in endothelial cells contribute significantly to overall VEGF-A dependent signaling in whole murine islets (*Figure 6—figure supplement 1F and G*).

Binding of VEGF-A to receptors triggers their endocytosis instantly, which is the initial step for subsequent intracellular signaling, and mainly regulated by EphrinB2 (*Sawamiphak et al., 2010*; *Wang et al., 2010*; *Nakayama et al., 2013*). Several studies have shown that VEGFR2 does not signal from the plasma membrane but from other intracellular compartments, mainly early endosomes and the endosomal sorting complex required for transport (ESCRT) (*Nakayama et al., 2013*; *Bruns et al., 2010*). The observed attenuation of VEGFR2 downstream signaling may stem from

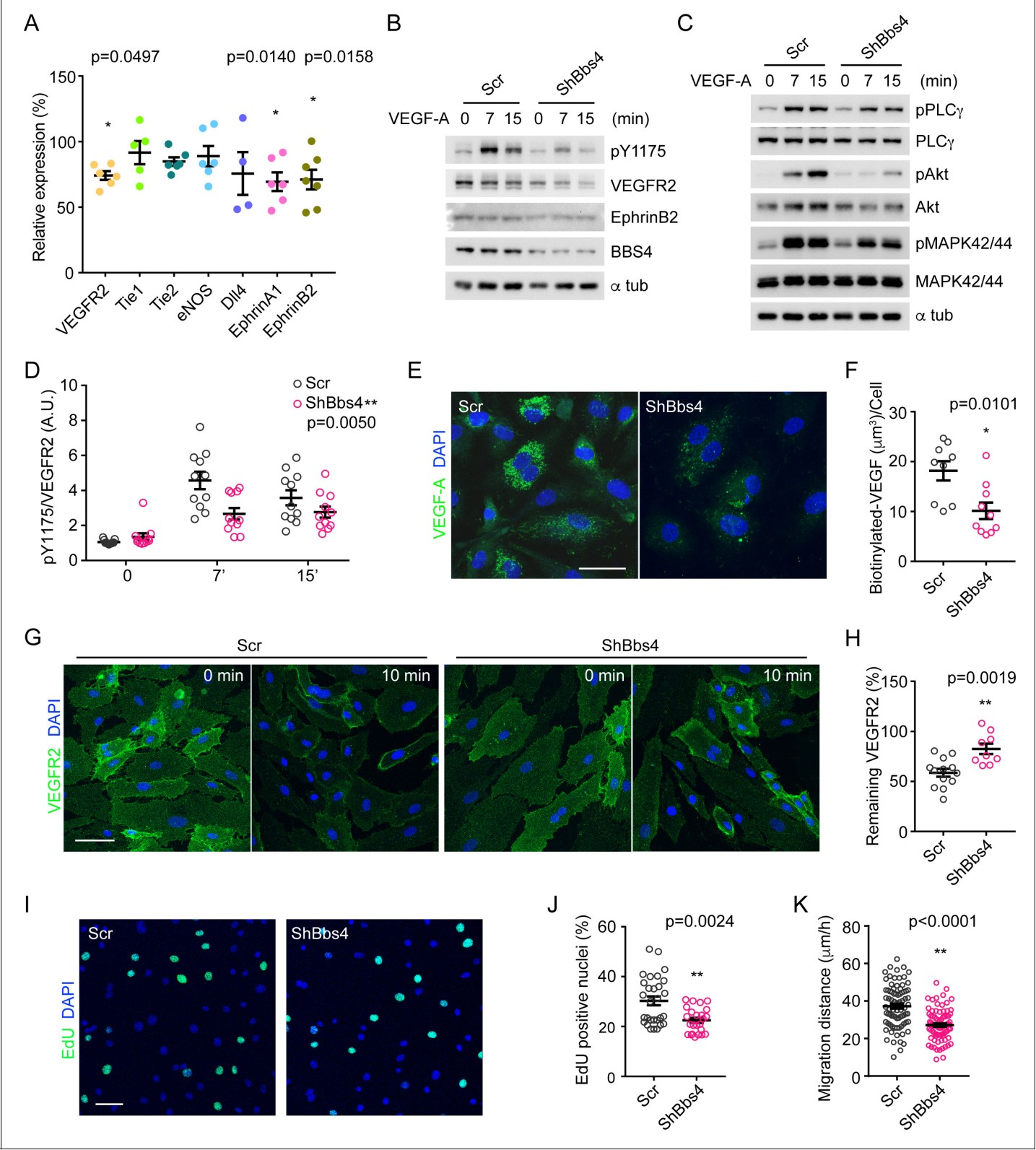

**Figure 6.** VEGFA-VEGFR2 induced signaling pathway and angiogenic response is disrupted in *Bbs4* silenced endothelial cells. (**A**) qPCR analysis of VEGF-A signaling related gene expression in *Bbs4* silenced MS-1. Results are presented as relative mRNA levels normalized to scrambled shRNA treated cells, *p<0.05, **p<0.01 by one-way-ANOVA, n = 3. (**B**) and (**C**) Western blots showing VEGF-A signaling pathway in scrambled or shBbs4 treated HDMECs. (**D**) Quantification of VEGFR2 activation. (**E**) Representative images showing biotinylated VEGF-A uptake in scrambled or ShBbs4

*Figure 6 continued on next page*

*Figure 6 continued*

treated HDMECs and quantification (**F**). (**G**) Representative cell surface staining of VEGFR2 in scrambled (Scr) or ShBbs4 treated HDMECs, prior to (0 min) and after VEGF-A stimulation (10 min). (**H**) Quantification of the percentage of remaining membrane-bound VEGFR2. (**I**) VEGF-A induced proliferation in Scr or ShBbs4 treated MS-1 by EdU assay (green) and quantification (**J**). (**K**) VEGF-A induced endothelial cell migration in Scr or ShBbs4 treated MS-1. Individual data points are shown, *p<0.05, **p<0.01 by Mann-Whitney test, n = 3. Scale bars: 50 μm.

The online version of this article includes the following figure supplement(s) for figure 6:

**Figure supplement 1.** VEGF-A gene expression, secretion and signaling in *Bbs4* or *Ift88* depleted cells.

dysregulation of these initial steps. Therefore, we examined the uptake of biotinylated VEGF-A in *BBS4*-depleted HDMECs. After 30 min incubation, *BBS4*-depleted cells had only internalized 45% VEGF-A compared to controls (*Figure 6E and F*, p=0.0101). We also tested VEGF-A uptake dynamics in a pancreatic endothelial cell line, MS-1 cells. Similar to our observations in HDMECs, control MS-1 cells efficiently and rapidly internalized VEGF-A, while by comparison, *Ift88*-depleted MS-1 cells internalized less VEGF-A after 30 min (61% compared to controls, *Figure 6—figure supplement 1H–1J*, p=0.0027). We further tested if VEGFR2 internalization was impaired in *BBS4*-depleted HDMECs (*Figure 6G*). Plasma membrane-bound VEGFR2 was evident prior to VEGF-A stimulation in both control and *BBS4*-depleted HDMECs. After ten min incubation with VEGF-A, a significant number of VEGFR2 was internalized in control cells, while 25% more VEGFR2 remained on the cell surface in *BBS4*-silenced cells, which indicates decreased internalization of VEGFR2 (*Figure 6H*, p=0.0019) and may be explained by our previous finding of lowered EphrinB2 levels (*Figure 6A and B*). These results suggest that the VEGFR2 internalization upon ligand binding is regulated by ciliary and basal body proteins.

VEGF-A signaling triggers a complex array of biological processes including endothelial cell proliferation and migration, two characteristic processes of angiogenesis. We tested the effect of *Bbs4*-depletion on proliferation by ethynyldeoxyuridine (EdU) incorporation in MS-1 cells. Under the chosen conditions, VEGF-A induced cell proliferation in *Bbs4*-depleted cells was reduced by 27% compared to controls cells (*Figure 6I and J*, p=0.0024). A scratch assay revealed that *Bbs4*-depleted cells covered only 73% of the distance of control cells in the same time frame (*Figure 6K*, p<0.0001), similar to previous reports of ciliary involvement in cell migration (*Vion et al., 2018*). These independent lines of evidence, i.e. reduced endothelial cell proliferation, migration, and impaired vessel fenestration, all point to a role for cilia and basal body proteins in VEGF-A/VEGFR2 dependent regulation of endothelial cell function.

An important negative regulator of VEGF-A dependent signaling is Notch1 –dependent signaling (*Williams et al., 2006*). Therefore, we tested the ability of Ift88 depleted pancreatic endothelial cells to respond to Notch1- dependent signaling (*Figure 6—figure supplement 1K*). Placing firefly luciferase under the control of multimerized CSL response elements upstream of a minimal promoter showed that the Notch1 intracellular domain (NICD) elicited a weaker response in *Ift88* depleted cells compared to controls. We did not observe different responses in the absence of NICD suggesting that the phenotype was specific to Notch1- dependent signaling. Because Notch1 signaling is a negative regulator of VEGF-A dependent signaling, loss of Notch1 signaling cannot explain the observed phenotype.

We also examined the relative expression of the same set of angiogenic factors in freshly isolated islets and found, while most expression levels remain unchanged, VEGF-A expression was reduced by 20% in *Bbs4*$^{-/-}$ islets compared to wt controls. *Von-Hippel-Lindau* gene expression, a known ciliary protein mutated in Von Hippel-Lindau disease and a known regulator of Hypoxia induced factor 1a, remains unchanged (*Figure 6—figure supplement 1L*). VEGF-A gene expression was also slightly decreased in βICKO islets (*Figure 6—figure supplement 1M*). To test if this reduction was physiologically relevant, we tested VEGF-A secretion in *Bbs4*$^{-/-}$ and βICKO islets. Loss of Bbs4 or Ift88 function respectively decreased VEGF-A secretion by 33% and 28% (*Figure 6—figure supplement 1N*, p=0.0063 and *Figure 6—figure supplement 1O*, p=0.0070), which could be explained by the reduction in VEGF-A gene expression. Altogether, these results consistently suggest that the intra-islet VEGF-A signaling pathway is impaired in *Bbs4*$^{-/-}$ islets.

## Discussion

Several recent studies have addressed the role of primary cilia in endothelial cell function (*Nauli et al., 2011*; *Kallakuri et al., 2015*; *Dinsmore and Reiter, 2016*; *Vion et al., 2018*). However, the precise role of primary cilia in islet vascularization remains unclear. Here, we reveal the role of the ciliary/basal body machinery in islet vascularization during islet islet engraftment upon transplantation. $Bbs4^{-/-}$ islets exhibited lower intra-islet capillary density and enlarged capillaries at two months of age. Similarly, $Pifo^{-/-}$ islets also have a reduced intra-islet capillary density and increased vessel diameter compared to littermate controls at the same age. This suggests that the observed defects are probably due to dysfunctional cilia and not related to other, non-ciliary functions of $Bbs4$. Of note, although Bbs4, Pifo and Ift88 are linked to ciliary function, their individual roles do not overlap precisely. Bbs4 localizes to the BBSome complex at the basal body and is involved in sorting of proteins to and from the ciliary compartment, but cilia still form. By comparison, Pifo mediates Aurora Kinase A (AurA)-dependent cilium disassembly resulting in persistent ciliation and bifurcation of cilia. In addition, cilia function is impaired as evidenced by laterality defects in $Pifo^{-/-}$ mice (*Kinzel et al., 2010*). By contrast, Ift88 depletion ablates all cilia. Importantly, though, vesicle docking still occurs (*Schmidt et al., 2012*), resulting in an abnormal cell that should not be interpreted as a wildtype, unciliated cell. Phenotypic variations could be explained by these different roles, but the fact that they are overlapping strongly implicates the cilia/basal machinery in endothelial function.

Previous work has shown hyperglycemia and elevated circulating insulin levels in $Bbs4^{-/-}$ mice at 4–6 months of age. This suggests increased insulin demand and thus insulin resistance, which could potentially affect islet vascularity. Indeed, studies with three independent insulin resistance mouse models showed that intra-islet vessel area significantly increased, primarily due to morphological changes, namely vessel dilation but not angiogenesis (*Dai et al., 2013*). Similarly, studies in Zucker diabetic fatty rats (ZDF) also demonstrated that early stages of insulin resistance and islet hyperplasia are accompanied by hypervascularization (*Li et al., 2006*). Of note, $Bbs4^{-/-}$ mice only become significantly overweight at 3–4 months of age, while we observe changes in islet vascularization at 2 months of age. We believe that we can exclude confounding effects of increasing body weight and insulin resistance on islet vascularization at the time of our investigation because the intra-islet capillary density is decreased, and not increased as is the case for models of insulin resistance.

Islet vascularization is typically regulated by paracrine signaling between endocrine cells (including α-, β- and δ-cells) and endothelial cells in the tissue surrounding the islet. Islet transplantation into the ACE and subsequent non-invasive and longitudinal monitoring of re-vascularization recapitulated this phenotype. In addition, this approach allows us to observe islets depleted of $Bbs4$ in a wildtype environment similar to targeted knockout islets. Given the problems with β-cell or islet specific Cre lines such as leakiness (*Liu et al., 2010*) or activity in the nervous system (*Honig et al., 2010*) our approach avoids these confounding factors. On the other hand, removing cilia from all endothelial cells also makes it difficult to exclude secondary, indirect effects that stem from endothelial cells in another tissue. In our experimental setup, both control and $Bbs4^{-/-}$ islets are transplanted into the eyes of the same wt animal, and vice versa, which minimizes the variability between different recipient animals and therefore makes it ideal for comparison.

Several lines of evidence suggest that the observed phenotype is largely driven by impaired ciliary and basal body function in endothelial cells. First, $Bbs4^{-/-}$ islets that were transplanted two days after isolation were re-vascularized at a lower rate than those transplanted after seven days. Prolonged islet cultivation depletes donor endothelial cells (in this case $Bbs4^{-/-}$ endothelial cells). $Bbs4^{-/-}$ islet endothelial cells are less efficient at islet revascularization compared to wt endothelial cells of the recipient. Second, wt islets transplanted into the ACE of $Bbs4^{-/-}$ mice were more slowly re-vascularized when transplanted after seven days compared to two days of tissue culture. The former condition shifts the composition of the islet vasculature toward a higher proportion of ciliary/basal body deficient $Bbs4^{-/-}$ endothelial cells, which are less responsive to the angiogenic demands than wt endothelial cells. Third, native and transplanted βICKO islets (β-cell-specific $Ift88$ depletion) show no obvious defect in capillary density, diameter or revascularization rate. Although we cannot exclude a minor contribution by reduced VEGF-A secretion from both $Bbs4^{-/-}$ and βICKO islets, the effect on islet vascularization is likely minimal and mostly compensated for by circulating VEGF-A. Islet vascularization is in part modulated by changing the actomyosin cortex of islet cells; while we have shown

EMT-like polarization defects in β-cells depleted of Ift88, we did not observe a complete loss of intra-islet capillary formation nor an enveloping structure of blood vessels surrounding the islets in βICKO mice (*Kragl et al., 2016*; *Volta et al., 2019*). Similarly, primary cilia are also present on some α- and δ-cells of the murine pancreatic islet (*Hughes et al., 2020*); we cannot fully exclude that they might play a role in islet re-vascularization, and that this might explain the difference between the findings in global knockout models *Bbs4*[-/-] and *Pifo*[-/-] or βICKO respectively. Because previous work has shown that β-cell VEGF-A secretion is responsible for the bulk of islet vascularization we believe that α-cell cilia are unlikely key denominators of islet vascularization. Here we suggest that there is an important, albeit not exclusive role for the endothelial ciliary/basal body apparatus in islet vascularization.

VEGF-A/VEGFR2 is a key signaling pathway for maintaining the especially dense and leaky islet microvasculature. Our results suggest that dysfunctional primary cilia alter the VEGF-A/VEGFR2 paracrine signaling activity in *Bbs4*[-/-] islets by impairing the internalization of ligand-bound VEGFR2 receptor. VEGF-A/VEGFR2 internalization and subsequent phosphorylation are disrupted in endothelial cells depleted of *BBS4* and *Ift88*, respectively. Several downstream signaling components are insufficiently activated in good agreement with previous observations. We conclude that defective VEGF-A/VEGFR2 dependent signaling in endothelial cells is the underlying cause of the observed phenotypes, namely reduced intra-islet capillary density and impaired fenestration of intra-islet micro-vessels. However, we cannot rule out a possible role of the Notch1 signaling pathway in the observed vascular phenotypes, although we did not detect significant changes in Dll4 expression in *Bbs4*[-/-] islets. Extensive studies are required to further pinpoint the involvement of other angiogenic signals which could also be affected by cilia dysfunction.

Intra-islet capillary networks provide endocrine cells with nutrients, oxygen and growth factors which support the development and maturation of islets. In addition, the microvasculature constantly adapts its barrier properties to the dynamic metabolic rates (*Pi et al., 2018*). Finely tuned interplay between nutrient delivery to islet endocrine cells, endocrine cell function and insulin disposal into the blood stream is essential for the maintenance of glucose homeostasis (*Brissova et al., 2006*; *Iwashita et al., 2007*). Whereas intra-islet capillary density normalizes over time in *Bbs4*[-/-] islets, vessel function remains impaired when cilia/basal function is compromised more than four months after transplantation. Blood vessel fenestration is significantly reduced in *Bbs4*[-/-] islets. Fenestrae are key features of islet capillaries that enable efficient nutrient delivery from and insulin disposal into blood vessels. In vivo 2-NBDG diffusion is less efficient and leakage of 40 KDa dextran diminished when *Bbs4* is deleted from intra-islet capillaries. With respect to glucose handling, decreased vessel permeability in *Bbs4*[-/-] islets could impose an additional barrier on insulin disposal, adding to the observed defects in 1[st] phase insulin secretion. We thus suggest that defective delivery of nutrients such as glucose to β-cells and the disposal of insulin from β-cells into the blood stream may contribute to impaired glucose handling of *Bbs4*[-/-] mice (*Gerdes et al., 2014*).

Functional imaging of micro-vessels remains challenging and is largely limited to easily accessible tissues such as cornea and skin (*Honkura et al., 2018*; *Wang et al., 2018*). Transplantation of islets into the ACE not only renders established intra-islet capillaries more accessible for standard imaging techniques but also provides the opportunity to follow dynamic processes such as islet re-vascularization under physiological as well as pathological conditions. This technique may also prove to be a useful resource for further investigations into blood vessel function in other tissues when transplanted into the ACE.

Finally, re-vascularization and establishing a functional interface between islet cells and blood supply are not only important in the context of islet transplantation but will likely prove vital for all approaches to β-cell replacement therapy. Moreover, considering the scarcity of organ donations worldwide, it is of paramount importance to optimize the outcomes of organ transplants. Revascularization rates in liver transplants correlate with long-term graft survival, renal function and early allograft dysfunction (*Buchholz et al., 2018*). Understanding the signaling role of primary cilia in this process provides us with novel targets to improve tissue re-vascularization and hence graft survival.

## Materials and methods

**Key resources table**

| Reagent type (species) or resource | Designation | Source or reference | Identifiers | Additional information |
|---|---|---|---|---|
| Cell line (*Homo sapiens*) | HEK 293FT | Thermo Fisher Scientific | # R70007 RRID:CVCL_6911 | Human embryonic kidney |
| Cell line (*Homo sapiens*) | HDMEC | Promocell | C-12212 | Primary human dermal microvascular endothelium |
| Cell line (*Mus musculus*) | MS-1 | ATCC | #CRL-2279 RRID:CVCL_D134 | Pancreas/islet of Langerhans; endothelium |
| Antibody | Goat polyclonal anti-PECAM-1 | R&D systems | AF3628 | (IF 1:400) |
| Antibody | Rabbit polyclonal anti-NG2 | Merck | AB5320 | (IF 1:200) |
| Antibody | Guinea pig polyclonal anti-Insulin | Dako | A0564 | (IF 1:1000) |
| Antibody | Rabbit polyclonal anti-BBS4 | Proteintech | 12766–1-AP | (WB 1:1000) |
| Antibody | Rabbit polyclonal anti-EphrinB2 | Thermo Fisher Scientific | BS-10659R | (WB 1:500) |
| Antibody | Rabbit monoclonal anti-phospho-VEGFR2 Y1175 | Cell Signaling Technology | 2478 | (WB 1:1000) |
| Antibody | Rabbit monoclonal anti-VEGFR2 | Cell Signaling Technology | 9698 | (IF 1:200) (WB 1:1000) |
| Antibody | Rabbit monoclonal anti-phospho-PLCγ1 Y783 | Cell Signaling Technology | 2821 | (WB 1:1000) |
| Antibody | Rabbit monoclonal anti-PLCγ1 | Cell Signaling Technology | 2822 | (WB 1:1000) |
| Antibody | Rabbit monoclonal anti-phospho-Akt Thr308/473 | Cell Signaling Technology | 4051 | (WB 1:1000) |
| Antibody | Rabbit monoclonal anti-pan Akt | Cell Signaling Technology | 4691 | (WB 1:1000) |
| Antibody | Rabbit monoclonal anti-phospho- p44/42 MAPK Thr202/Tyr204 | Cell Signaling Technology | 4370 | (WB 1:2000) |
| Antibody | Rabbit monoclonal anti- p44/42 MAPK | Cell Signaling Technology | 4695 | (WB 1:2000) |
| Antibody | Mouse monoclonal anti-α-tubulin | Sigma-Aldrich | T8203 | (WB 1:2000) |
| Antibody | Donkey monoclonal anti-Rabbit IgG, peroxidase-linked species-specific whole antibody Secondary Antibody | GE Healthcare | 10794347 | (WB 1:6000) |

*Continued on next page*

*Continued*

| Reagent type (species) or resource | Designation | Source or reference | Identifiers | Additional information |
|---|---|---|---|---|
| Antibody | Sheep monoclonal anti-mouse IgG, peroxidase-linked species-specific whole antibody Secondary Antibody | GE Healthcare | 10196124 | (WB 1:6000) |
| Peptide, recombinant protein | Recombinant Human VEGF 165 | Peprotech | 100–20 | 100 µg/ml in 1XPBS |
| Peptide, recombinant protein | Recombinant Human VEGF 165, Biotinylated Protein | R&D systems | BT293 | 100 µg/ml in 1XPBS |
| Commercial assay or kit | Mouse VEGF Quantikine ELISA Kit | R&D systems | MMV00 | |
| Commercial assay or kit | PowerUp SYBR Green Master Mix | Thermo Fisher Scientific | A25742 | |
| Commercial assay or kit | Dextran, Texas Red, 70,000 MW, Neutral | Thermo Fisher Scientific | D1830 | 5 mg/ml in 1X PBS |
| Commercial assay or kit | Dextran, Fluorescein, 40,000 MW, Anionic | Thermo Fisher Scientific | D1844 | 2.5 mg/ml in 1X PBS |
| Commercial assay or kit | Click-iT EdU Cell Proliferation Kit for Imaging, Alexa Fluor 488 dye | Thermo Fisher Scientific | C10337 | |
| Chemical compound, drug | 2-NBDG (2-(N-(7-Nitrobenz-2-oxa-1,3-diazol-4-yl)Amino)—2-Deoxyglucose) | Thermo Fisher Scientific | N13195 | 5 mg/ml in 1X PBS |
| Chemical compound, drug | Streptozotocin | Sigma-Aldrich | S0130-50MG | |
| Software, algorithm | MATLAB | MathWorks | RRID:SCR_001622 | R2016a |
| Software, algorithm | Volocity | PerkinElmer (Massachusetts, USA) | RRID:SCR_002668 | |
| Software, algorithm | Fiji-ImageJ | http://imagej.net/Fiji | RRID:SCR_002285 | |

## Animal models

Experimental procedures involving live animals were carried out in accordance with animal welfare regulations and with approval of the Regierung Oberbayern (Az 55.2-1-54-2532-187-15 and ROB-55.2–2532.Vet_02-14-157) or in accordance with the Karolinska Institutet's guidelines for the care and use of animals in research, and were approved by the institute's Animal Ethics Committee respectively (Ethical permit number 19462–2017). C57Bl/6J B6 albino and C57Bl/6J mice were obtained from The Jackson Laboratory (Maine, USA). Ai14 mice and *Cdh5-Cre* mice were obtained from the Jackson laboratory (stock number 007914 and 017968). *Bbs4*[-/-] mice were generated in the Lupski lab and backcrossed to C57Bl/6 J (*Eichers et al., 2006*). Heterozygous offspring were crossed to produce homozygous *Bbs4*[-/-] and littermate control. *Pifo*[wt/-] mice were crossed to produce homozygous *Pifo*[-/-] mice and littermate controls (*Kinzel et al., 2010*). βICKO animals were generated by

crossing *Pdx1-CreER* mice (*Zhang et al., 2005*) with *Ift88^loxP/loxP* mice (*Volta et al., 2019*; *Davenport et al., 2007*).

## Islet isolation and transplantation

Islets were isolated by cannulation of the common bile duct of donor animals and infusion of 2.5 ml collagenase P, 1.0 mg/ml in HBSS containing 25 mM HEPES and 0.2% BSA (Sigma-Aldrich, USA). Inflated pancreata were dissected out and digested at 37°C for 10 min, and cultured in RPMI 1640 Medium (11 mM D-glucose) supplemented with 10% fetal bovine serum, 100 IU/ml penicillin, 100 µg/ml streptomycin and 2 mM L-Glutamine (all from Thermo Fisher Scientific, USA). Transplanted islets in all experimental setups have an average diameter of 200 µm.

4-month-old female *Bbs4^-/-* mice and their wildtype littermates were used as islet donors. 3–4 month-old female mice were used as recipients in all the in vivo transplantations except for *Bbs4^-/-* animals (*Figures 3D–3G* and *4* males and one female knockout animals were used as recipients). In all experiment setups, wt and *Bbs4^-/-* islets were transplanted into each eye of the same recipient mouse to minimize the variations caused by different recipients. Similarly, wt islets cultivated for different periods were transplanted into each eye of the same *Bbs4^-/-* recipient for comparison. For transplantation, recipient mice were anesthetized with 2% isoflurane (Baxter, USA) and fixed with a custom-made head holder (Narishige, Japan) as described previously (*Speier et al., 2008*). A small incision was made in the cornea with a 25 G needle and a glass cannula containing islets was inserted through the opening into the anterior chamber of the eye. Around 4–8 islets were carefully positioned on the iris around the pupil. Post-operative analgesia was done by subcutaneous injection of 2 µg Temgesic (RB Pharmaceuticals Limited, UK).

## Streptozotocin treatment and glucose tolerance test

For Streptozotocin (STZ, Sigma-Aldrich, USA) treatment, 4-month-old female C57BL/6J B6 albino mice were fasted for 6 hr and a single dose of 200 mg STZ per kg body weight in phosphate buffered saline (PBS, Thermo Fisher Scientific, USA) was injected intraperitoneally. For glucose tolerance tests, animals were fasted for 6 hr and 2 g D-glucose (20% solution in PBS+/+) per kg body weight was injected intraperitoneally. Blood glucose values were measured afterwards at specified time points using an Accu-Chek Aviva system (Roche, Switzerland).

## In vivo imaging of islet grafts

Imaging was performed between one to twelve weeks post-transplantation by confocal microscopy. Images of islet grafts were obtained by a Leica SP5 system with 25 × objective (N.A.0.95, Leica Microsystems, Germany). Viscotears (Laboratoires Théa, France) was used as an immersion medium between the lens and the mouse eyes. For visualization of islet vascularization, animals were anesthetized with isoflurane (Baxter, USA). 100 µl of PBS solution containing 5 mg/ml of 70 kDa Texas Red-conjugated dextran (Thermo Fisher Scientific, USA) was injected intravenously prior to imaging. Z-stacks of 2 µm thickness were acquired for every islet graft (Ex.: 561 and 594 nm, Em.: 558–564 nm and 600–700 nm). For quantification of vascular leakage, animals were anesthetized with a mixture of 1:1 Hypnorm and midazolam (Roche, Switzerland). 100 µl of PBS solution containing 2.5 mg/ml of 40 kDa FITC-conjugated dextran (Thermo Fisher Scientific, USA) was injected intravenously prior to imaging. Time series of z-stacks (2 µm) were acquired every second (Ex.: 488 and 633 nm, Em.: 500–550 nm and 630–636 nm). For monitoring the uptake of glucose analog, animals were anesthetized with a mixture of 1:1 Hypnorm and midazolam (Roche, Switzerland). 2-(N-(7-Nitrobenz-2-oxa-1,3-diazol-4-yl)Amino)−2-Deoxyglucose (2-NBDG, Thermo Fisher Scientific, USA) was dissolved in PBS at 5 mg/ml. Recipient mice were anesthetized and a tail vein catheter containing 50 µl 2-NBDG solution was installed. Time series of z-stacks (2 µm) were acquired every second (Ex.: 488 and 633 nm, Em.: 500–600 nm and 630–636 nm) and 2-NBDG solution was injected during imaging.

## Image analysis

Unprocessed original images were used in all quantifications of in vivo re-vascularization. Graft volume was estimated from backscatter signals using Volocity image analysis software (Perkin Elmer, USA). Dextran-labeled islet capillaries were identified at different time points, using the same thresholds for all groups at all times, and structures smaller than 10 µm$^3$ were automatically excluded.

Their diameters and volumes were calculated using Volocity. Re-vascularization rate at 1 to 8 weeks post-transplantation was calculated as the percentage of islet vascular volume at each individual time point to that of the same islet at 12 week end point. Relative vascularity of each islet graft at 12 weeks post-transplantation was calculated from dividing the volume of islet vessels by the estimated graft volume. For FITC-dextran and 2-NBDG leakage analysis, images were first movement corrected in MATLAB (MathWorks, USA). Afterwards, time lapse confocal image stacks were analyzed and fluorescence intensity was quantified using Fiji (*Schindelin et al., 2012*) with the plugin Time Series Analyzer V3. Islet cells were identified by backscatter signals. The simulation of in vivo leakage kinetics was done in GraphPad Prism using the model of plateau followed by one phase association.

Image analysis for histological sections were performed with Volocity. The region of interest per stack (ROI) was drawn according to insulin staining. The surface area and volume of PECAM-1, NG2, and insulin-positive structures were calculated using Volocity. Islet capillary density was obtained by counting the PECAM-1-positive volume and dividing it by the insulin-positive volume enclosed in this region. Pericyte coverage was calculated by counting NG2-positive surface area and dividing it by PECAM-1-positive surface area in the same insulin-positive region. Vessel diameters were taken from 6 to 10 randomly chosen segments per islet.

## Cell culture

HEK 293FT cells were purchased from Thermo Fisher Scientific (USA). MS-1 cells were purchased from ATCC (USA). Human Dermal Microvascular Endothelial Cells (HDMECs) were purchased from PromoCell (Germany). HEK293FT cells were cultured in Dulbecco's Modified Eagle's Medium (DMEM) supplemented with 2 mM L-Glutamine and 10% fetal bovine serum (all from Thermo fisher Scientific, USA). MS-1 cells were cultured in Dulbecco's Modified Eagle's Medium (DMEM) supplemented with 2 mM L-Glutamine and 5% fetal bovine serum. HDMECs were cultured in Endothelial Cell Growth Medium MV2 with Endothelial Cell Growth Supplement (PromoCell, Germany). All cells have been tested negative for mycoplasma contamination. The identities of HEK 293FT and MS-1 haven't been re-authenticated, but they have been used within limited passages upon purchasing.

## Construction of lentiviral vectors

We used a commercial lentiviral system from Sigma-Aldrich for designing and construct shRNA containing lentiviral vectors. Two verified sequences (*Mus musculus*) and one verified sequence (*Homo sapiens*) per gene of interest (*Mus musculus*) were selected from MISSION*shRNA* database (https://www.sigmaaldrich.com/life-science/functional-genomics-and-rnai/shrna/individual-genes.html). For *Bbs4*: 5'- CCTTGTATTAAGAACCTAG-3' (shBbs4-1) and 5'-GCATGACCTGACTTACATAAT-3' (shBbs4-2). For *Ift88*: 5'-TCAGATGCCATCAACTCATTT-3' (shIft88-1) and 5'-CTATGAGTCATACAGG TATTT-3' (shIft88-2). For human *Bbs4*: 5'-TAGTCCTCAGAGTGCTGATAA-3'. Control scrambled sequence: 5'-CCTAAGGTTAAGTCGCCCTCG-3'. The corresponding oligonucleotides were cloned into pLKO.1-puro vector (Sigma-Aldrich, USA) according to manufacturer's instructions. The efficacy was assessed by relative level of message RNA by qPCR. For reasons of clarity, we refer to more effective shRNA construct shBbs4-1 as shBbs4 and shIft88-1 as shIft88 in the text.

## Lentiviral production and endothelial cell transduction

Lentivirus was produced in 293FT cells and harvested following the manufacturer's guidelines (https://www.sigmaaldrich.com/life-science/functional-genomics-and-rnai/shrna/trc-shrna-products/lentiviral-packaging-mix.html). $1 \times 10^6$ MS-1 or $2 \times 10^5$ HDMEC cells were plated in 25 cm$^2$ flasks and 1 ml of crude virus containing medium was used for transduction. Transduced cells were further screened in 1.0 µg/ml puromycin (Sigma-Aldrich, USA) for a week before use.

## VEGF-A uptake

$1 \times 10^5$ transduced HDMECs or $5 \times 10^5$ MS-1 cells were plated on coverslips in six well plates. Confluent cells were starved in DMEM supplemented with 0.5% BSA for 6 hr, and incubated with 50 ng/ml biotinylated VEGF-A (R&D Systems, USA) for 30 min at 37°C. Treated cells were subsequently fixed in 4% paraformaldehyde solution at room temperature for 15 min and permeabilized in 0.1% Triton-X (Sigma-Aldrich, USA) in PBS for 10 min. Fixed cells were then washed three times in PBS and blocked with 2% BSA in PBS for 1 hr. To detect the internalized biotinylated VEGF-A, cells were

further incubated with Streptavidin- FITC conjugate (1:1000, Thermo Fisher Scientific, USA) at room temperature for 1 hr.

## VEGFR-2 internalization

$1 \times 10^5$ transduced HDMECs were plated on coverslips in six well plates. Confluent cells were starved overnight in DMEM supplemented with 1% BSA. Starved cells were gently washed twice with PBS, and stimulated with 50 ng/ml human VEGF-A (Peprotech, Sweden) for 10 min at 37°C. VEGF-A containing medium was quickly removed and cells were washed twice in ice-cold PBS. Subsequently, cells were fixed in 4% paraformaldehyde solution at room temperature for 15 min, and blocked with 2% BSA in PBS for 1 hr without permeabilization. Further incubation steps with primary and secondary antibodies were described as below.

## VEGF-A ELISA

Isolated islets from control and *Bbs4*$^{-/-}$ animals were cultured overnight. 30 islets from each group were picked into four well plates containing 450 μl culture medium per well. 72 hr later, supernatants were collected and stored at −80°C. Islets were collected and lysed in 50 μl M-PER buffer (Thermo Fisher Scientific, USA), for quantification of DNA contents by Quant-iT PicoGreen dsDNA Assay Kit (Thermo Fisher Scientific, USA) later according to manufacturer's instructions. Secreted VEGF-A in the supernatants was measured with mouse VEGF Quantikine ELISA Kit (R&D Systems, USA), according to manufacturer's instructions.

## EdU labeling

Cell proliferation was detected with Click-iT EdU Alexa Fluor 488 Imaging Kit (Thermo Fisher Scientific, USA). Briefly, $2 \times 10^5$ transduced MS-1 cells were plated on coverslips in six well plates. After attachment, cells were starved overnight in DMEM supplemented with 1% BSA, gently washed twice with PBS, and further incubated with 50 ng/ml mouse VEGF-A (Peprotech, Sweden) and 10 μM EdU for 4 hr. Treated cells were fixed in 4% paraformaldehyde solution at room temperature for 15 min and EdU detection was carried out according to manufacturer's instructions.

## Endothelial cell migration assay

$2.5 \times 10^5$ transduced MS-1 cells were plated in 12 well plates. Confluent cells were washed gently with PBS, and incubated with 50 ng/ml VEGF-A in DMEM supplemented with 1% BSA. A scratch was made with 1 ml tip in the middle of each well, cells were washed again with PBS, and images were acquired for documentation of the initial distances between the separated cells. 18 hr later, cells were fixed in 4% paraformaldehyde solution at room temperature for 15 min, and the same areas were photographed again for calculating the distances of migration in individual wells. Images were acquired with BD Pathway 855 Bioimaging Systems using a Olympus 10x objective (BD, USA).

## Notch1 activity assay

$3 \times 10^4$ transduced MS-1 cells were plated in 96 well plates for transfection of Notch1 reporter. Notch1 signaling pathway activity was detected according to manufacturer's instructions (BPS Bioscience, #60509).

## Transmission electron microscopy (TEM)

Islet grafts were dissected out from euthanized animals which have been transplanted previously, and peri-islet iris tissues were removed carefully. Grafts were fixed in 2.5% glutaraldehyde+1% paraformaldehyde in 0.1 M phosphate buffer, pH 7.4 at 4°C overnight. Samples were further washed in 0.1 M phosphate buffer, pH 7.4 and post-fixed in 2% osmium tetroxide 0.1 M phosphate buffer, pH 7.4 at 4°C for 2 hr, dehydrated in ethanol and acetone, and embedded in LX-112 (Ladd Research Industries, Burlington, USA). Ultra-thin sections of 50–60 nm were cut by Leica EM UC 6 (Leica Microsystems, Germany) and contrasted with uranyl acetate, followed by lead citrate treatment and checked in Tecnai 12 Spirit Bio TWIN transmission electron microscope (FEI company, The Netherlands) at 100 kV. Digital images were acquired by Veleta camera (Olympus Soft Imaging Solutions, Germany).

## Pancreatic sections and immunofluorescent staining

Pancreata were swiftly removed from euthanized animals, rinsed in PBS and fixed in 4% paraformaldehyde in PBS at room temperature for 2 hr. Tissues were then washed with PBS twice and transferred stepwise to 10%, 20% and eventually 30% sucrose in PBS. Tissues were then embedded in O.C.T. freezing medium (Thermo Fisher Scientific, USA) at −80°C and subsequently sectioned into 20 µm thick sections.

Cells and islets were fixed in 4% paraformaldehyde solution at room temperature for 15 min or 1 hr, followed by permeabilization in 0.1% Triton-X in PBS, and blocking with 2% BSA or 10% goat serum in PBS respectively. Samples were incubated with primary antibodies, including anti-PEACAM-1 antibody (goat, 1:400, R&D Systems, USA), purified rat anti-mouse CD31 antibody (rat, 1:100, BD Bioscience, USA), anti-NG2 antibody (rabbit, 1:100, Millipore, USA), anti-insulin antibody (guinea pig, 1:1000, Dako, Sweden) and anti-VEGFR2 N-terminal antibody (rabbit, 1:100, R&D Systems, USA), at room temperature for 1 hr or at 4°C overnight. After washing with PBS, cells and tissues were further incubated with secondary antibodies (goat anti-rabbit IgG H+L Alexa Fluor 633, goat anti mouse IgG H+L Alexa Fluor 488 or 633, goat anti guinea pig IgG H+L Alexa Fluor 561 and donkey-anti goat IgG H+L 488; all at 1:1000, Thermo Fisher Scientific, USA) at room temperature for 1 hr, and mounted with ProLong Antifade mountant with DAPI (Thermo Fisher Scientific, USA).

## Western blots

Confluent HDMECs were stimulated by 50 ng/ml VEGF-A for indicated lengths of time and lysed in modified RIPA buffer (150 mM NaCl, 50 mM Tris-HCl pH 7.4, 1% NP-40, 0.1% sodium deoxycholate, 1 mM EDTA, all from Sigma-Aldrich, USA) supplemented with protease and phosphatase inhibitors (Mini Complete/PhosphoStop, Roche Diagnostics, Germany). Protein contents were determined using the BCA Protein Assay Kit (Pierce, USA). Protein extracts were separated on 7.5% stain-free precast gels (Bio-Rad, USA) and transferred to 0.22 µm PVDF membranes (GE Healthcare, USA). Blots were probed with primary antibodies, including phospho-VEGFR2 Y1175 antibody (rabbit, 1:1000), VEGFR2 antibody (rabbit, 1:1000), EphrinB2 antibody (rabbit, 1:500), phospho-PLCγ1 Y783 antibody (rabbit, 1:1000), PLCγ1 antibody (rabbit, 1:1000), phospho-Akt Thr308/473 antibody (rabbit, 1:1000), Akt1/2 antibody (rabbit, 1:1000), phospho-p44/42 MAPK Thr202/Tyr204 antibody (rabbit, 1:2000) and Erk1/2 antibody (rabbit, 1:2000; all from Cell Signaling Technology, USA), and α-tubulin antibody (mouse,1:2000, Sigma-Aldrich, USA), and subsequently with secondary anti-mouse/-rabbit IgG (H+L)-HRP conjugates (1:6000, GE Healthcare, USA). Quantification was carried out on original scanned images using Image Lab software (Bio-Rad, USA) and normalization to total protein.

## Quantitative RT-PCR

Total RNA samples were collected from isolated islets of control and *Bbs4*⁻ᐟ⁻ animals, or transduced MS-1 cells. Tissue and cell samples were lysed and total RNA was isolated using the GeneJET RNA purification kit (Thermo Fisher Scientific, USA) according to manufacturer's instructions. Extracted mRNA was transcribed into cDNA using the Maxima First Strand cDNA Synthesis Kit (Thermo Fisher Scientific, USA) following manufacturer's instructions. *Bbs4, Ift88, VEGF-A, VEGFR2, Ang-1, Ang-2, Tie-1, Tie-2, VHL, EphrinA1, EphrinB2, eNOS* and *Dll4* messages were measured using the SYBR Green Master Mix (Thermo Fisher Scientific, USA) with an QuantStudio 5 Real-Time PCR System (Applied Biosystems, USA). Relative gene expression level was quantified against the average of *18S*, *HMBS* and *TBP* messages, and normalized to those of control cells. All primer sequences are listed in *Table 1*.

## Statistical tests

All results are presented as either original data points or mean ± SEM. Measurements and calculated results obtained from the WT and control groups were subjected to normality test, and most of them pass Kolmogorov-Smirnov test. Two-way ANOVA test was used for all time course analysis, One-way ANOVA test was used for comparison of multiple groups, and Mann-Whitney test was used to assess statistical significance for other experiments, with a $p$ value < 0.05 considered to indicate significance (GraphPad Prism).

**Table 1.** Sequences of qPCR primers.

| Gene | Sequence |
| --- | --- |
| 18S | L:5'-AACCCGTTGAACCCCATT-3' R:5'-CCATCCAATCGGTAGTAGCG-3' |
| HMBS | L:5'-CGGAGTCATGTCCGGTAAC-3' R:5'-GGTGCCCACTCGAATCAC-3' |
| TBP | L:5'-TGCTGTTGGTGATTGTTGGT-3' R:5'-CTGGCTTGTGTGGGAAAGAT-3' |
| Bbs4 | L:5'-AATGCACTGACCTACGACCC-3' R:5'-ATGCTGATGCATACTGCTGC-3' |
| Ift88 | L:5'-TGAAGTGGCAGCTGATGGTA-3' R:5'-CTGTGCAGAGACGAACCAAG-3' |
| VEGF-A | L:5'-CAGGCTGCTGTAACGATGAA-3' R:5'-GCATTCACATCTGCTGTGCT-3' |
| VEGFR2 | L:5'-GGCGGTGGTGACAGTATCTT-3' R:5'-GTCACTGACAGAGGCGATGA-3' |
| Ang-1 | L:5'-GATCTTACACGGTGCCGATT-3' R:5'-TTAGATTGGAAGGGCCACAG-3' |
| Ang-2 | L:5'-TCCAAGAGCTCGGTTGCTAT-3' R:5'-AGTTGGGGAAGGTCAGTGTG-3' |
| Tie-1 | L:5'-TCAACTGCAGCTCCAAAATG-3' R:5'-TGACAGCTCTGTCCAAAACG-3' |
| Tie-2 | L:5'-GTGTGAGAAAGAAGGCAGGC-3' R:5'-GTAGGTAGTGGCCACCCAGA-3' |
| VHL | L:5'-CAGGAGACTGGACATCGTCA-3' R:5'-TCCTCTTCCAGGTGCTGACT-3' |
| EphrinA1 | L:5'-CCCACATTACGAGGACGACT-3' R:5'-GTGAAGCGCTGGAATTTCTC-3' |
| EphrinB2 | L:5'-AGGAATCACGGTCCAACAAG-3' R:5'-GTCTCCTGCGGTACTTGAGC-3' |
| eNOS | L:5'-GCAAGACCTCCTGAGGACAG-3' R:5'-TGCAAAGAAAAGCTCTGGGT-3' |
| Dll4 | L:5'-CAGAGACTTCGCCAGGAAAC-3' R:5'-TCATTTTGCTCGTCTGTTCG-3' |

# Acknowledgements

We thank Professor Kari Alitalo for helpful discussions and kindly providing AAVs encoding VEGF-A for in vitro testing in isolated islets, Kjell Hultenby at Karolinska Institutet's core facility for electron microscopy for obtaining the TEM images, and Dr. Stefan Jacob for help of image analysis. This study was supported by DZD funding and a Marie-Curie International Re-integration grant (JMG), the Swedish Research Council, the Novo Nordisk Foundation, Karolinska Institutet, the Swedish Diabetes Association, The Family Knut and Alice Wallenberg Foundation, Diabetes Research and Wellness Foundation, Swedish Foundation for Strategic Research, Berth von Kantzow's Foundation, The Skandia Insurance Company Ltd., Strategic Research Programme in Diabetes at Karolinska Institutet, ERC-2013-AdG 338936-BetaImage, the Stichting af Jochnick Foundation and the Family Erling-Persson Foundation.

# Additional information

### Funding

| Funder | Grant reference number | Author |
| --- | --- | --- |
| German Center for Diabetes Research | | Jantje Mareike Gerdes |
| FP7 People: Marie-Curie Actions | International Reintegration Grant PIRG07-GA-2010-268397 | Jantje Mareike Gerdes |
| Swedish Research Council | | Per-Olof Berggren |
| Novo Nordisk Fonden | | Per-Olof Berggren |
| Karolinska Institutet | | Yan Xiong |
| Swedish Strategic Research Program Diabetes | | Per-Olof Berggren |
| Swedish Diabetes Association | | Per-Olof Berggren |
| Knut and Alice Wallenberg Foundation | | Per-Olof Berggren |

| | | |
|---|---|---|
| Diabetes Research & Wellness Foundation | | Per-Olof Berggren |
| Berth von Kantzows Stiftelse | | Per-Olof Berggren |
| Skandia Insurance Company Ltd | | Per-Olof Berggren |
| European Research Council | ERC-2018-AdG 834860 EYELETS | Per-Olof Berggren |

The funders had no role in study design, data collection and interpretation, or the decision to submit the work for publication.

### Author contributions
Yan Xiong, Conceptualization, Data curation, Formal analysis, Validation, Investigation, Visualization, Methodology, Writing - original draft, Writing - review and editing; M Julia Scerbo, Francesco Volta, Formal analysis, Investigation; Anett Seelig, Investigation; Nils O'Brien, Formal analysis; Andrea Dicker, Resources, Investigation; Daniela Padula, Heiko Lickert, Resources; Jantje Mareike Gerdes, Conceptualization, Data curation, Supervision, Funding acquisition, Investigation, Methodology, Writing - original draft, Writing - review and editing; Per-Olof Berggren, Conceptualization, Writing - review and editing

### Author ORCIDs
Yan Xiong https://orcid.org/0000-0003-2339-130X
Jantje Mareike Gerdes https://orcid.org/0000-0001-6885-5441

### Ethics
Animal experimentation: This study was performed in strict accordance to the German and Swedish animal welfare legislation. Experimental procedures involving live animals were carried out in accordance with animal welfare regulations and with approval of the Regierung Oberbayern (Az 55.2-1-54-2532-187-15 and ROB-55.2-2532.Vet_02-14-157) or in accordance with the Karolinska Institutet's guidelines for the care and use of animals in research, and were approved by the institute's Animal Ethics Committee respectively (Ethical permit number 19462-2017).

### Decision letter and Author response
Decision letter https://doi.org/10.7554/eLife.56914.sa1
Author response https://doi.org/10.7554/eLife.56914.sa2

## Additional files
### Supplementary files
• Transparent reporting form

### Data availability
All data generated or analyzed during this study are included in this manuscript and supporting files.

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
