## [Decision Letter]

**Acceptance summary:**

Through the use of mouse models as well as cell-based approaches this study demonstrates that primary cilia on endothelial cells regulate pancreatic islet vascularization and vascular barrier function via the VEGF-A/VEGFR2 signaling pathway. The work improves our understanding of how cilia regulate islet function and whole body glucose homeostasis, and may help explain why mutations in ciliary genes lead to metabolic disorders and obesity.

**Decision letter after peer review:**

Thank you for submitting your article "Islet vascularization is regulated by primary endothelial cilia via VEGF-A dependent signaling" for consideration by *eLife*. Your article has been reviewed by three peer reviewers, and the evaluation has been overseen by a Reviewing Editor and Piali Sengupta as the Senior Editor. The following individuals involved in review of your submission have agreed to reveal their identity: Eckhard Lammert (Reviewer #1).

The reviewers have discussed the reviews with one another and the Reviewing Editor has drafted this decision to help you prepare a revised submission.

Summary:

The study sets out to uncover the mechanism underlying islet primary cilia control of islet function and in turn whole-body glucose homeostasis. The authors had previously shown that primary cilia on islet β-cells regulate insulin secretion. Observed changes in islet vascular density in *Bbs4^-/-^* mice, supported by well-planned re-vascularization experiments of *Bbs4^-/-^* islet transplanted into WT mice, and vice versa, showed that islet primary cilia also do play some role in supporting vascularization. These observations were corroborated in another ciliopathy model (Pifo KO) further implicating primary cilia as being important to neo-vascularization. Cell-specific primary cilia contribution to islet vascularization was also addressed to a certain extent by showing that deletion of primary cilia on β-cells did not affect islet vascularization. Through in vitro as well as ex vivo studies, the authors concluded that primary cilia on endothelial cells, and separately islet primary cilia, facilitate VEGFR2 signaling required for normal neo-vascular growth. in vitro studies are thorough in their design and in their use of experimental controls. The authors should be commended for putting together this large body of work and for presenting it in an elegant, easy to understand, manner.

Revisions:

1) The authors do not cite sufficient original work (such as the first paper describing the role of VEGF in islet vascularisation and glucose tolerance, published in Curr Biology 2003, or the paper on the biomechanics of islets in vessel development, published in Nat Comms 2016. They should introduce these papers from the beginning to better highlight why they designed their experiments as they did.

2) While use of multiple knockout models is a positive, it is likely that each affects cilia and the role of cilia in islet vascularization in different ways. Please discuss.

3) It remains unclear whether primary endothelial cell cilia are solely responsible for the observed islet phenotype, as stated in the title. Cilia were suggested to be on other cell-types (eg. α-cells, Hughes et al., 2020). Specific deletion of primary endothelial cell cilia would have greatly helped address the hypothesis. The authors state that endothelial cell-specific cilia KO from other tissues could potentially confound the interpretation of the data. This is a valid concern and the work could be strengthened by showing that primary cilia were only found on endothelial and β-cells (and this can be shown through primary cilia distribution within the islet). The βICKO mouse results sufficiently addresses the concern of β-cell contribution (with the caveat of the slight time gap of 8wk in *Bbs4^-/-^* and approx 12wk in βICKO).

4) Figure 6—figure supplement 1D. To pinpoint endothelial cell culpability in the observed changes, it will be useful to test whether VEGF-A still stimulates downstream phosphorylation cascade in endothelial-cell depleted islets (after 7 days).

5) It would be helpful to the reader to introduce the rationale, advantages and disadvantages for the different models. For example, introducing the *Bbs4^-/-^* mouse model before detailing the experimental results would help the reader.

6) In Figure 1A, is islet mass affected in knockout mice at two months and further at four months of age? Does variation in islet size directly affect the vessel density measurements?

7) Bbs4 knockout mice have increased body weight. Are these mice generally healthy? Does knocking out Bbs4 lead to peripheral insulin resistance and islet hypertrophy and hyperplasia (as in other models of insulin resistance)? Please discuss whether the increase in body weight would affect the islet response to insulin resistance and vascularization.

8) Please discuss how the current results relate to prior reports from some of the authors on the manuscript: a) Sci Rep. 2016 Feb 22;6:21448. doi: 10.1038/srep21448. Non-invasive cell type selective in vivo monitoring of insulin resistance dynamics. b) FASEB J. 2019 Jan;33(1):204-218. doi: 10.1096/fj.201800826R. Diet-induced β-cell insulin resistance results in reversible loss of functional β-cell mass.

9) What is the contribution of donor and recipient endothelial cell contribution during re-vascularization in these models (as related to Nyqvist et al., 2005)?

10) Does the size of transplanted islets affect the re-vascularization rate? Were similar islet equivalents transplanted into the eye each time?

11) Please provide additional rationale for the transition to cell lines to study molecular mechanisms and impaired VEGFA-VEGFR2 dependent signaling (Figure 6). Please discuss how the in vitro cell studies relate to overall process addressed in Figures 1-5 in primary tissue and cells (and the data related to this in supplemental figures).

12) An explanation of the engraftment timeline or a schematic showing your various timepoints would help the reader.

13) Please clarify these experimental details:

a) Does PECAM-1 expression differ between knockout and control mice?

b) Please discuss why there is a plateau after dextran infusion. Is the eye filling with dextrans at a certain point and not allowing further movement of dextrans?

c) Was the reader of the fenestration measurements blinded to the samples?

d) To what was the fluorescence in Figure 5E normalized? If normalized to islet size, were similarly sized islets used for these measurements?

e) In Figure 5H, β cells are not specifically labeled, but the y-axis indicates this.

---

## [Author Response]

Revisions:1) The authors do not cite sufficient original work (such as the first paper describing the role of VEGF in islet vascularisation and glucose tolerance, published in Curr Biology 2003, or the paper on the biomechanics of islets in vessel development, published in Nat Comms 2016. They should introduce these papers from the beginning to better highlight why they designed their experiments as they did.

Our sincerest apologies for not sufficiently citing the work that this study is built on. We are standing on the shoulders of giants, indeed, and have failed to acknowledge it.

While islet vessel formation itself does not depend on islet derived VEGF-A, fenestration is strictly dependent on VEGF-A expression in islet cells and VEGF-A dependent signaling in endothelial cells. In addition, biomechanical properties of islet cells direct a sorting process of endothelial and epithelial cells and ultimately switch between invasion into or envelopment of islet cells by endothelial cells. We have shown that β-cell polarity is disturbed when cilia are impaired and that this is also affecting actin dynamics. However, *Bbs4^-/-^*islets are vascularized, albeit at a lower density than those of littermates, suggesting that the change in biomechanical properties of Bbs4 depleted cells does not lead to a complete switch towards vessels enveloping the islets.

We have changed the manuscript as follows:

“While pancreas-specific VEGF-A ablation does not completely block islet vascularization, islet fenestration is crucially dependent on VEGF-A/ VEGFR2-dependent signaling. Delivery of nutrients to β-cells and disposal of insulin into the blood vessels is modulated by fenestrated blood vessels that are fine tuning the system to optimize glucose tolerance.”

“Changing the biomechanical properties such as actomyosin cortex tension of epithelial cells in developing islets by deletion of Integrated-linked kinase (ILK) specifically blocks intra-islet capillary formation completely and leads to glucose intolerance. Instead, blood vessels accumulate in the islet periphery and form an envelope-like structure suggesting that biomechanics of epithelial tissues are important determinants of blood vessel formation.”

We have also added to the Discussion:

“Islet vascularization is in part modulated by changing the actomyosin cortex of islet cells; while we have shown EMT-like polarization defects in β-cells depleted of Ift88, we did not observe a complete loss of intra-islet capillary formation nor an enveloping structure of blood vessels surrounding the islets in βICKO mice.”

Again, we apologize for our omission and hope that our revisions have amended for our mistake.

2) While use of multiple knockout models is a positive, it is likely that each affects cilia and the role of cilia in islet vascularization in different ways. Please discuss.

Thank you for your comment. Indeed, we feel that the use of multiple knockout models can help us triangulate the role primary cilia play in the process of islet vascularization and nutrient/ insulin delivery to/ from the blood stream.

*Bbs4^-/-^* mice are a model of Bardet-Biedl Syndrome (BBS) and show reasonable overlap with the symptoms that BBS patients manifest. While ciliogenesis is not interrupted in these animals, cilia are not fully functional. The BBSome is thought to mediate sorting of membrane associated proteins to and from the cilium. Some evidence suggests that cells are not polarized correctly in Bbs-depleted cells and non-canonical, β-catenin independent signaling (planar cell polarity signaling) is impaired^1-4^. We have shown that EMT-like polarity defects lead to defects in actin polymerization dynamics and early endocytosis.

*Pifo*^-/-^ mice show defects in cilia disassembly and re-entry into the cell cycle mediated by AurA activity. PIFO accumulates at the basal body during cilia disassembly and interacts with ciliary targeting complexes. While the exact role of PIFO in cilia function is not known, known defects include left-right asymmetry defects and heart looping defects. This suggests that cilia function is impaired and that at least a subset of ciliary signaling pathways- such as the symmetry breaking morphogen gradient- are affected by cilia dysfunction.

Depletion of Ift88 from cells leads to defects in ciliogenesis and cilia maintenance and function. Importantly, the early steps of ciliogenesis including vesicle docking are still taking place, so Ift88-depleted cells are not simply cells without a cilium, they are dysfunctional. We have shown that Ift88 depletion leads to stabilization of β-catenin in the cell and subsequent EMT-like polarity defects in an insulinoma cell line. We suggest that similar impairments are present in endothelial cells and that this is affecting actin dynamics.

We have added a paragraph to the Discussion:

“Of note, although Bbs4, Pifo and Ift88 are linked to ciliary function, their individual roles do not overlap precisely. Bbs4 localizes to the BBSome complex at the basal body and is involved in sorting of proteins to and from the ciliary compartment, but cilia still form. By comparison, Pifo mediates Aurora Kinase A (AurA)-dependent cilium disassembly resulting in persistent ciliation and bifurcation of cilia. In addition, cilia function is impaired as evidenced by laterality defects in *Pifo*^-/-^ mice^5^. By contrast, Ift88 depletion ablates all cilia. Importantly, though, vesicle docking still occurs, resulting in an abnormal cell that should not be interpreted as a wildtype, unciliated cell. Phenotypic variations could be explained by these different roles, but the fact that they are overlapping strongly implicates the cilia/ basal machinery in endothelial function.”

3) It remains unclear whether primary endothelial cell cilia are solely responsible for the observed islet phenotype, as stated in the title. Cilia were suggested to be on other cell-types (eg. α-cells, Hughes et al., 2020). Specific deletion of primary endothelial cell cilia would have greatly helped address the hypothesis. The authors state that endothelial cell-specific cilia KO from other tissues could potentially confound the interpretation of the data. This is a valid concern and the work could be strengthened by showing that primary cilia were only found on endothelial and β-cells (and this can be shown through primary cilia distribution within the islet). The βICKO mouse results sufficiently addresses the concern of β-cell contribution (with the caveat of the slight time gap of 8wk in Bbs4^-/-^ and approx 12wk in βICKO).

Thank you for your thoughtful comment. We can confirm that α- but also a subset of δ- cells are ciliated as well. However, in a β-cell-specific VEGF-A knockout mouse model, islet vascularization is strongly impacted- although α-cells still express VEGF-A. While we cannot fully exclude a role for α- and δ-cell cilia in the vascularization process, we think that it is unlikely that they are playing a major role in the islet vascularization process. We have added the following paragraph to the Discussion:

“Similarly, primary cilia are also present on some α- and δ-cells of the murine pancreatic islet^6^; we cannot fully exclude that they might play a role in islet re-vascularization, and that this might explain the difference between the findings in global knockout models *Bbs4*^-/-^ and *Pifo*^-/-^ or βICKO respectively. Because previous work has shown that β-cell VEGF-A secretion is responsible for the bulk of islet vascularization we believe that α-cell cilia are unlikely key denominators of islet vascularization. Here we suggest that there is an important, albeit not exclusive role for the endothelial ciliary/ basal body apparatus in islet vascularization.”

4) Figure 6—figure supplement 1D. To pinpoint endothelial cell culpability in the observed changes, it will be useful to test whether VEGF-A still stimulates downstream phosphorylation cascade in endothelial-cell depleted islets (after 7 days).

Thank you for this suggestion. To test whether VEGF-A stimulation persists in islets depleted of endothelial cells, we isolated islets from 2 female C57/BL6 mice and kept them in culture for 7 days. On the day of experiment, islets were starved for 1.5 hrs and then incubated in control buffer or VEGF-A containing buffer for 10 minutes for stimulation, similar to what has been done with freshly isolated islets. Afterwards, protein lysates were used in Western blots for the examination of VEGF-A downstream signaling. As shown in Figure 6—figure supplement 1, VEGF-A stimulated phosphorylation in Akt and MAPK 42/44 is only minimal in this case. We have added the panels to Figure 6—figure supplement 1F and 1G, and revised the section as follows:

“Importantly, similar experiments on wt islets depleted of intra-islet endothelial cells show markedly lower response to VEGF-A stimulation; therefore, VEGF-A/ VEGFR2 dependent signaling cascades in endothelial cells are major contributors to overall VEGF-A dependent signaling in intact murine islets (Figure 6—figure supplement 1F and 1G)”

5) It would be helpful to the reader to introduce the rationale, advantages and disadvantages for the different models. For example, introducing the Bbs4^-/-^ mouse model before detailing the experimental results would help the reader.

We apologize for the omission and have added a short Introduction accordingly:

“*Bbs4^-/-^* mice are a model of Bardet-Biedl Syndrome (OMIM #209900), a ciliopathy characterized mainly by polydactyly, renal and gonadal malformations and truncal obesity. *Bbs4* encodes a protein that is a component of the BBSome complex, a protein complex thought to be involved in the sorting of membrane proteins to and from the ciliary compartment^1,2,7^. While *Bbs4*^-/-^ mice form primary cilia, they are not fully functional. The symptoms show good overlap with what has been observed in BBS patients and include obesity, male sterility, and impaired glucose handling.”

“*Pitchfork* (*Pifo*^-/-^) mice have dysfunctional cilia that cannot be properly disassembled^5^. PIFO does not localize to the ciliary axoneme but accumulates at the base of the cilium during cilia disassembly and interacts with ciliary targeting complexes. Laterality defects in *Pifo*^-/-^ mice suggest that ciliary signaling is impaired in these animals.”

“Ift88 is essential to ciliary maintenance and formation, and cells depleted of Ift88 do not have ciliary axonemes, although the early stages of ciliogenesis, vesicle docking to centrosomes, are still complete.”

6) In Figure 1A, is islet mass affected in knockout mice at two months and further at four months of age? Does variation in islet size directly affect the vessel density measurements?

Thank you for raising these valid concerns. We have previously published that islet size distribution was slightly skewed towards larger islets in 1-month-old male *Bbs4*^−/−^ mice without clear differences in range, β-cell mass and pancreatic insulin content are not significantly different from those of littermates^7^. We have not tested β-cell mass at later stages but could isolate islets from 4 months old animals, suggesting that no overt loss of β-cell mass occurs. To address your concern, we have performed additional immunostaining on pancreatic sections from 2- and 4-month-old wt and *Bbs4^-/-^* mice respectively, and correlation between vessel density and β-cell volume (estimated by insulin staining) was calculated in each group as shown in Author response image 1. As the Pearson coefficient suggests, there is no correlation between vascular density and β-cell mass.

7) Bbs4 knockout mice have increased body weight. Are these mice generally healthy? Does knocking out Bbs4 lead to peripheral insulin resistance and islet hypertrophy and hyperplasia (as in other models of insulin resistance)? Please discuss whether the increase in body weight would affect the islet response to insulin resistance and vascularization.

Thank you for raising this issue. Because models of obesity and peripheral insulin resistance show hypervascularization in the early stages of islet cell hyperplasia, we do not think that there is a confounding effect of insulin resistance on islet vascularization in *Bbs4*^-/-^*mice*. Interestingly, vessel diameter could be related to insulin resistance although the findings are not clear.

We have added a statement as follows:

“Previous work has shown hyperglycemia and elevated circulating insulin levels in *Bbs4*^-/-^ mice at 4-6 months of age. This suggests increased insulin demand and a certain level of insulin resistance, which could potentially affect islet vascularity. Indeed, studies with three independent insulin resistance mouse models showed that intra-islet vessel area significantly increased, primarily due to morphological changes, namely vessel dilation but not angiogenesis. Similarly, studies in Zucker diabetic fatty rats (ZDF) also demonstrated that early stages of insulin resistance and islet hyperplasia are accompanied by hypervascularization. Of note, *Bbs4*^-/-^ mice only become significantly overweight at 3-4 months of age, while we observe changes in islet vascularization at 2 months of age. We believe that we can exclude confounding effects of increasing body weight and insulin resistance on islet vascularization at the time of our investigation because the intra-islet capillary density is decreased, and not increased as is the case for models of insulin resistance.”

8) Please discuss how the current results relate to prior reports from some of the authors on the manuscript: a) Sci Rep. 2016 Feb 22;6:21448. doi: 10.1038/srep21448. Non-invasive cell type selective in vivo monitoring of insulin resistance dynamics. b) FASEB J. 2019 Jan;33(1):204-218. doi: 10.1096/fj.201800826R. Diet-induced β-cell insulin resistance results in reversible loss of functional β-cell mass.

We thank the reviewers for this comment. In the two studies, we generated biosensors for the detection of insulin resistance and functional islet mass. We took advantage of the optical properties of the eye and the in vivo imaging platform to longitudinally monitor Western diet induced changes in β-cell functionality. In our study, we used fluorescent dyes to track the newly formed vessels during islet engraftment and evaluate the barrier function of islet vessels. While there is some evidence that *Bbs4*^-/-^ mice are hyperinsulinemic, their obesity phenotype is related to hyperphagia of standard chow and not high fat/ high sucrose diet. We transplanted *Bbs4*^-/-^ islets into wt mice without an additional dietary challenge, and transplanting wt islets into *Bbs4*^-/-^ mice did not suggest additional effects by persistent insulin resistance (Figure 3D,E). While we agree that the question of hyperinsulinemia, insulin signaling and insulin resistance in ciliopathy models such as *Bbs4^-/-^* is important and warrants further study, we feel that it is beyond the scope of this manuscript.

9) What is the contribution of donor and recipient endothelial cell contribution during re-vascularization in these models (as related to Nyqvist et al., 2005)?

We thank the reviewers for this question and apologize for the confusion. In Nyqvist et al., 2005, Tie2-GFP islets were used for transplantation labelling all the remaining endothelial cells in donor islets. In this study, donor islet endothelial cells form vessel structures in the islet graft when transplanted shortly after isolation. However, the exact contribution of donor endothelial cells to the newly formed graft vessels was not quantified. In our current study, we have used a different model and approach for discerning the contribution from the two cell populations, as shown in Figure 3—figure supplement 1. We transplanted wt and *Bbs4*^-/-^ islets (after 2 days in culture) into *Cdh5-tdT* mouse eyes, which means that all recipient endothelial cells express tdTomato. Cells in the newly formed vessels that are negative for tdTomato are derived from donor islets. We quantified the surface area ratio of tdTomato positive structures to the FITC-dextran labeled vascular structures, as a measure of contribution from recipient endothelial cells. This ratio shifts slightly during islet engraftment, and by the 12-wk time point it’s on average 41.7% in wt and 45.9% in Bbs4^-/-^ islets. One caveat of our method is, although the labelling efficiency/specificity of tdTomato in *Cdh5-tdT* mice is high, it may not be 100% in all the animals. In addition, the accuracy of the ratio calculation also depends on imaging depth of the lasers used for detecting FITC-dextran and tdTomato in vivo. We would like to point out that this result is consistent with another study where freshly picked islets were transplanted under the kidney capsule, and the donor endothelial cells identified by immunofluorescent staining constitute 40±3% of the newly formed vasculature at 3-5 weeks post-transplantation (Brissova et al., 2004).

10) Does the size of transplanted islets affect the re-vascularization rate? Were similar islet equivalents transplanted into the eye each time?

Thank you for these questions. It has been shown in multiple studies that islets are exposed to different levels of hypoxia when they are cut off from the blood stream and transplanted to a new site^8,9^. This leads to the increased expression and activation of various hypoxia induced angiogenic factors, including VEGF-A, which may result in different re-vascularization rates between small and large islets^10^.

To avoid confounding effects, we have consistently used similar-sized islets in all transplantations and re-vascularization analysis. The islets that we have chosen had an average diameter of 200 µm.

11) Please provide additional rationale for the transition to cell lines to study molecular mechanisms and impaired VEGFA-VEGFR2 dependent signaling (Figure 6). Please discuss how the in vitro cell studies relate to overall process addressed in Figures 1-5 in primary tissue and cells (and the data related to this in supplemental figures).

We apologize if this section was somewhat confusing and felt unrelated to the previous sections. The advantage of cell lines are the supply of material and the relative ease by which pure endothelial cellular material can be obtained. In addition to murine pancreatic endothelial cell line MS-1, we have also used Human Dermal Microvascular Endothelial Cells (HDMEC), to validate our experimental findings in human primary cells. As mentioned before, vascular fenestration of intra-islet capillaries is causally linked to VEGF-A/ VEGFR2-dependent signaling. Our findings suggest that in Bbs4- and Ift88-depleted cells VEGFR2, EphrinA1 and B2 are downregulated, VEGF-A uptake and VEGFR2 internalization are reduced. Because VEGFR2 signaling activity does not only occur from the plasma membrane but also depends on EphrinB2 mediated internalization, these lines of evidence suggest that this pathway is impaired in endothelial cells void of functioning cilia. This is supported by the fact that VEGFR2 phosphorylation and phospho-activation of downstream signaling nodes is decreased in Bbs-4 depleted endothelial cells.

We have changed the text as follows:

“We have observed a role for endothelial cilia in the islet vascularization process; the signaling pathways involved in this process, however, remained elusive. To complement the in vivo data and to better understand the cell-autonomous role of ciliary/basal body proteins in endothelial cells, we used a murine pancreatic endothelial cell line, MS^-1^. Suppressing *Bbs4* by RNA interference reduces *Bbs4* expression up to two-fold (Figure 6—figure supplement 1A). Expression analysis of key angiogenic factors revealed no observable changes in Tie1, Tie2, endothelial Nitric oxide synthase (eNOS) or Notch ligand Δ-4 (Dll4) (Figure 6A). However, VEGFR2, Ephrin A1 and B2 are downregulated by 20% when MS^-1^ cells are depleted of Bbs4. Interestingly, Bbs4 depletion has no effect on PECAM-1 expression and therefore does not change characteristic properties of this endothelial cell line. Of note, intra-islet capillary density and fenestration is primarily maintained by VEGF-VEGFR2-signaling^11-13^.”

12) An explanation of the engraftment timeline or a schematic showing your various timepoints would help the reader.

We thank the reviewers for this comment. We have included a schematic depicting the differences between the two transplantation strategies and added the timeline of in vivo experiments in Figure 1—figure supplement 2B.

Briefly, islets were isolated from donor animals, and cultured for 2 or 7 days, before they were transplanted into recipient animals. Islet engraftment and re-vascularization was examined by in vivo imaging at 1, 2, 3, 4, 6, 8 and 12 weeks post-transplantation, respectively.

13) Please clarify these experimental details:a) Does PECAM-1 expression differ between knockout and control mice?

We have not been able to determine examined PECAM-1 expression levels in control and *Bbs4*^-/-^ islet endothelial cells; because vessel density is lower in *Bbs4*^-/-^ islets compared to controls at an early age, gene expression analysis will be skewed. However, we have compared PECAM-1 expression level in Scr and ShBbs4 treated MS^-1^ cells, which is not significantly different. The result is shown in Author response image 2 as relative gene expression levels in ShBbs4 treated MS^-1^ cells, normalized to that of Scr cells.

**Author response image 2. respfig2:** 

b) Please discuss why there is a plateau after dextran infusion. Is the eye filling with dextrans at a certain point and not allowing further movement of dextrans?

There are several factors contributing to the plateau of dextran fluorescence. First, the animals were given a single bolus injection of dextran at the 1-min time point, and the total amount of dextran in circulation soon declines after injection, due to clearance by the liver and kidney of recipient animal. Consequently, dextran diffusion from the blood flow into the aqueous humor slows down as the concentration gradient becomes smaller. Meanwhile, there is constant flow of the aqueous humor, and the average turnover rate is 2.5% per minute^14^. This further hinders the accumulation of dextran in aqueous humor after the initial peak following the intravenous injection.

c) Was the reader of the fenestration measurements blinded to the samples?

Yes, the reader of the fenestration measurements is blinded to all the samples. We have added a clarifying statement in the manuscript.

d) To what was the fluorescence in Figure 5E normalized? If normalized to islet size, were similarly sized islets used for these measurements?

In our initial analysis, dextran fluorescence was not normalized. However, we used similar-sized islets in both groups for this experiment, and each islet has a diameter between 180-220 µm. We have now re-analysed the raw data, and normalized dextran fluorescence to islet size (with an average diameter of 200 µm). We have updated the panel and *p* value in Figure 5E and 5F. Thank you again for this suggestion.

e) In Figure 5H, β cells are not specifically labeled, but the y-axis indicates this.

We apologize for this mistake and confusion. We quantified average 2-NBDG fluorescence in all islet cells in our analysis, not specifically in β-cells. We have corrected this both in Figure 5H and the text.

**References**

1 Nachury, M. V. et al. A core complex of BBS proteins cooperates with the GTPase Rab8 to promote ciliary membrane biogenesis. Cell 129, 1201-1213, doi:10.1016/j.cell.2007.03.053 (2007).

2 Liew, G. M. et al. The intraflagellar transport protein IFT27 promotes BBSome exit from cilia through the GTPase ARL6/BBS3. Dev Cell 31, 265-278, doi:10.1016/j.devcel.2014.09.004 (2014).

3 Gerdes, J. M. et al. Disruption of the basal body compromises proteasomal function and perturbs intracellular Wnt response. Nat Genet 39, 1350-1360, doi:10.1038/ng.2007.12 (2007).

4 Simons, M. et al. Inversin, the gene product mutated in nephronophthisis type II, functions as a molecular switch between Wnt signaling pathways. Nat Genet 37, 537-543, doi:10.1038/ng1552 (2005).

5 Kinzel, D. et al. Pitchfork regulates primary cilia disassembly and left-right asymmetry. Dev Cell 19, 66-77, doi:10.1016/j.devcel.2010.06.005 (2010).

6 Hughes, J. W. et al. Primary cilia control glucose homeostasis via islet paracrine interactions. Proc Natl Acad Sci U S A 117, 8912-8923, doi:10.1073/pnas.2001936117 (2020).

7 Gerdes, J. M. et al. Ciliary dysfunction impairs beta-cell insulin secretion and promotes development of type 2 diabetes in rodents. Nat Commun 5, 5308, doi:10.1038/ncomms6308 (2014).

8 Olsson, R., Olerud, J., Pettersson, U. & Carlsson, P. O. Increased numbers of low-oxygenated pancreatic islets after intraportal islet transplantation. Diabetes 60, 2350-2353, doi:10.2337/db09-0490 (2011).

9 Lehmann, R. et al. Superiority of small islets in human islet transplantation. Diabetes 56, 594-603, doi:10.2337/db06-0779 (2007).

10 Kampf, C., Mattsson, G. & Carlsson, P. O. Size-dependent revascularization of transplanted pancreatic islets. Cell Transplant 15, 205-209, doi:10.3727/000000006783982124 (2006).

11 Esser, S. et al. Vascular endothelial growth factor induces endothelial fenestrations in vitro. J Cell Biol 140, 947-959, doi:10.1083/jcb.140.4.947 (1998).

12 Roberts, W. G. & Palade, G. E. Increased microvascular permeability and endothelial fenestration induced by vascular endothelial growth factor. J Cell Sci 108 ( Pt 6), 2369-2379 (1995).

13 Lammert, E. et al. Role of VEGF-A in vascularization of pancreatic islets. Curr Biol 13, 1070-1074, doi:10.1016/s0960-9822(03)00378-6 (2003).

14 Aihara, M., Lindsey, J. D. & Weinreb, R. N. Aqueous humor dynamics in mice. Invest Ophthalmol Vis Sci 44, 5168-5173, doi:10.1167/iovs.03-0504 (2003).